

# Physical and mechanical rock properties of a heterogeneous volcano; the case of Mount Unzen, Japan

Jackie E. Kendrick[1,2], Lauren N. Schaefer[3,4], Jenny Schauroth[1], Andrew F. Bell[2], Oliver D. Lamb[1,5], Anthony Lamur[1], Takahiro Miwa[6], Rebecca Coats[1], Yan Lavallée[1] and Ben M. Kennedy[3]

[1]Department of Earth, Ocean and Ecological Sciences, University of Liverpool, Liverpool, L69 3GP, UK.

[2]School of Geosciences, University of Edinburgh, James Hutton Road, Edinburgh, EH9 3FE, UK

[3]Department of Geological Science, University of Canterbury, Christchurch, 8140, New Zealand.

[4]Now at, U.S. Geological Survey, 1711 Illinois St, Golden, Colorado, 80401, USA

[5]Department of Geological Sciences, University of North Carolina at Chapel Hill, Chapel Hill, NC, USA

[6]National research institute for earth science and disaster resilience (NIED), Ibaraki, 305 0006, Japan

*Correspondence to*: Jackie E. Kendrick (Jackie.Kendrick@ed.ac.uk)

**Abstract.** Volcanoes represent one of the most critical geological settings for hazard modelling due to their propensity to both unpredictably erupt and collapse, even in times of quiescence. Volcanoes are heterogeneous at multiple scales, from porosity which is variably distributed and frequently anisotropic to strata that are laterally discontinuous and commonly pierced by fractures and faults. Due to variable and, at times, intense stress and strain conditions during and post-emplacement, volcanic

rocks span an exceptionally wide range of physical and mechanical properties. Understanding the constituent materials' attributes is key to improving the interpretation of hazards posed by the diverse array of volcanic complexes. Here, we examine the spectrum of physical and mechanical properties presented by a single dome-forming eruption at a dacitic volcano, Mount Unzen (Japan) by testing a number of isotropic and anisotropic lavas in tension and compression and using monitored acoustic emission (AE) analysis. The lava dome was erupted as a series of 13 lobes between 1991-1995, and its ongoing instability

means much of the volcano and its surroundings remain within an exclusion zone today. During a field campaign in 2015, we selected 4 representative blocks as the focus of this study. The core samples from each block span range in porosity from 9.14 to 42.81 %, and permeability ranges from $1.54 \times 10^{-14}$ to $2.67 \times 10^{-10}$ m$^2$ (from 1065 measurements). For a given porosity, sample permeability varies by > 2 orders of magnitude is lower for macroscopically anisotropic samples than isotropic samples of similar porosity. An additional 379 permeability measurements on planar block surfaces ranged from $1.90 \times 10^{-15}$ to 2.58

$\times 10^{-12}$ m$^2$, with a single block having higher standard deviation and coefficient of variation than a single core. Permeability under confined conditions showed that the lowest permeability samples, whose porosity largely comprises microfractures, are most sensitive to effective pressure. The permeability measurements highlight the importance of both scale and confinement conditions in the description of permeability. The uniaxial compressive strength (UCS) ranges from 13.48 to 47.80 MPa, and tensile strength (UTS) using the Brazilian disc method ranges from 1.30 to 3.70 MPa, with crack-dominated lavas being weaker

than vesicle-dominated materials of equivalent porosity. UCS is lower in saturated conditions, whilst the impact of saturation



on UTS is variable. UCS is between 6.8 and 17.3 times higher than UTS, with anisotropic samples forming each end member. The Young's modulus of dry samples ranges from 4.49 to 21.59 GPa and is systematically reduced in water-saturated tests. The interrelation of porosity, UCS, UTS and Young's modulus was modelled with good replication of the data. Acceleration of monitored acoustic emission (AE) rates during deformation was assessed by fitting Poisson point process models in a

Bayesian framework. An exponential acceleration model closely replicated the tensile strength tests, whilst compressive tests tended to have relatively high early rates of AEs, suggesting failure forecast may be more accurate in tensile regimes, though with shorter warning times. The Gutenberg-Richter $b$-value has a negative correlation with connected porosity for both UCS and UTS tests which we attribute to different stress intensities caused by differing pore networks. $b$-value is higher for UTS than UCS, and typically decreases (positive $\Delta b$) during tests, with the exception of cataclastic samples in compression. $\Delta b$

correlates positively with connected porosity in compression, and negatively in tension. $\Delta b$ using a fixed sampling length may be a more useful metric for monitoring changes in activity at volcanoes than $b$-value with an arbitrary starting point. Using coda wave interferometry (CWI) we identify velocity reductions during mechanical testing in compression and tension, the magnitude of which is greater in more porous samples in UTS but independent of porosity in UCS, and which scales to both $b$-value and $\Delta b$. Yet, saturation obscures velocity changes caused by evolving material properties, which could mask damage

accrual or source migration in water-rich environments such as volcanoes. The results of this study highlight that heterogeneity and anisotropy within a single system not only add uncertainty but also have a defining role in the channelling of fluid flow and localisation of strain that dictate a volcano's hazards and the geophysical indicators we use to interpret them.

## 1 Introduction

### 1.1 Rock Failure and Volcano Stability

Volcanoes are constructed over relatively short geological timescales via the accrual of diverse eruptive products that span a porosity range from 0 – 97 %, making them inherently unstable structures prone to collapse (Reid et al., 2000; McGuire, 1996; Delaney, 1992). Volcanoes experience deformation due to ongoing magmatic activity (Donnadieu et al., 2001; Voight et al., 1983), pore-fluid pressurisation thanks to active hydrothermal systems and regional tectonics including stress rotation (Reid

et al., 2010; Patanè et al., 1994), and alteration due to percolation of fluids (Rosas-Carbajal et al., 2016) and contact with intrusive bodies (Saubin et al., 2019; Weaver et al., 2020). In particular, volcanoes are often located in seismically active regions and may be susceptible to earthquake triggering (Walter et al., 2007; Surono et al., 2012). The presence of thermally-liable subvolcanic basement rocks (e.g. Mollo et al., 2011), or volcaniclastics (Cecchi et al., 2004) may enhance gravitational spreading (Borgia et al., 1992; van Wyk de Vries and Francis, 1997) that also increase instability. Large-scale heterogeneities

such as lithological contacts, unconsolidated layers, laterally discontinuous beds, as well as faults, including previous edifice collapse scars, also contribute to the propensity for volcanic edifices to collapse (e.g. Williams et al., 2019; Tibaldi, 2001; Carrasco-Núñez et al., 2006; Schaefer et al., 2019).



Lava domes may be particularly susceptible to collapse events both during and after emplacement. During emplacement the
development of gas overpressure, gravitational loading, uneven underlying topography, variations in extrusion direction and
intense rainfall can all trigger partial to complete dome collapse (Harnett et al., 2019b; Calder et al., 2002; Elsworth et al.,
2004; Voight and Elsworth, 2000). Once activity subsides lava domes are still prone to collapse due to fracturing induced by
contraction of cooling magma bodies (Lamur et al., 2018; Fink and Anderson, 2000), fault systems (Walter et al., 2015),
redistribution of mass post-emplacement (Elsworth and Voight, 1996) and hydrothermal alteration (Ball et al., 2015; Horwell
et al., 2013).

A primary control on volcano and lava dome stability are the physical and mechanical properties of the constituent materials.
Volcanic rocks host void space that ranges from vesicles with complex geometries to networks of elongate cracks or fractures
(e.g. Schaefer et al., 2015; Shields et al., 2016; Colombier et al., 2017; Herd and Pinkerton, 1997), and dome lavas in particular
frequently have anisotropic pore networks (Heap et al., 2014b; Lavallée and Kendrick, 2020). As porosity is the major control
on the strength and geophysical characteristics of geomaterials, such diversity of porosity translates to a broad spectrum of
mechanical behaviour of dome rocks and lavas (e.g. Harnett et al., 2019a; Heap et al., 2016a; Coats et al., 2018; Lavallée and
Kendrick, 2020), and a universal predictor of material strength eludes us. A key parameter in the description of lavas and
volcanic rock properties is permeability, which defines materials' ability to build and alleviate pore pressure, important during
eruptive activity and quiescence alike (Day, 1996; Saar and Manga, 1999; Mueller et al., 2005; Collinson and Neuberg, 2012;
Farquharson et al., 2015; Scheu et al., 2006b). Permeability of volcanic rocks has been shown to span some 10 orders of
magnitude, including as much as 5 orders of magnitude for materials of a given porosity (e.g. Saar and Manga, 1999; Mueller
et al., 2005; Farquharson et al., 2015; Klug and Cashman, 1996). Permeability, controlled by porosity, acts in competition with
material strength to define the fragmentation threshold, the limit over which pore pressure exceeds the material's strength and
triggers wholesale failure for a spectrum of porous geomaterials (Mueller et al., 2008; Spieler et al., 2004; Kremers et al., 2010;
Alatorre-Ibargüengoitia et al., 2010; Scheu et al., 2006b). This interplay influences both pressurised magmas and fluid-
saturated volcanic rocks, shifting the stress fields that may trigger failure (e.g. Voight and Elsworth, 2000), a consideration
pertinent to the interpretation of secondary hazards in scenarios of rapid pressurisation during tectonic earthquakes (e.g. Walter
et al., 2007) and during decompression induced by unloading during collapse (e.g. Hunt et al., 2018; Maccaferri et al., 2017;
Williams et al., 2019; Brantley and Scott, 1993).

Increasingly sophisticated numerical models have been utilised to interpret the conditions leading to partial or extensive
collapse of lava domes (e.g. Harnett et al., 2018; Sato et al., 1992; Voight and Elsworth, 2000), though such simulations
necessarily entail estimates for parameters such as dome structure, vent geometry and slope of substrata, and are dependent
upon accurate characterisation of physical and mechanical properties. Creating homogeneous zones, and assigning fixed values
or ranges of parameters for the purpose of isolating the influence of variables during modelling is commonplace and
computationally beneficial, yet this remains a great source of uncertainty by failing to account for the spectrum of dome



materials' properties. In the last few decades, a surge in laboratory testing means the characterisation of hot lavas and volcanic rocks has improved significantly and reliable constraints of rheological and mechanical properties are being obtained.

Rheology of natural lavas including those with suspended vesicles and crystals have been defined across a broad range of temperatures and rates using concentric cylinder and parallel plate methods (Cordonnier et al., 2009; Lavallée et al., 2007; Coats et al., 2018; Webb, 1997; Okumura et al., 2010; Chevrel et al., 2015; Kolzenburg et al., 2016). Volcanic rock strength inversely correlates with porosity, and is frequently defined in terms of uniaxial compressive strength (UCS) at room or high temperature (e.g. Heap et al., 2014b, c; Schaefer et al., 2015; Coats et al., 2018; Bubeck et al., 2017; Pappalardo et al., 2017),

direct and indirect tensile strength at room or high temperature (Harnett et al., 2019a; Lamur et al., 2018; Hornby et al., 2019; Lamb et al., 2017; Benson et al., 2012) and triaxial tests at varying pressures, temperatures and saturation conditions (Heap et al., 2016a; Smith et al., 2011; Farquharson et al., 2016; Shimada, 1986; Kennedy et al., 2009; Mordensky et al., 2019). Strength of volcanic rocks also typically positively correlates with strain rate (Schaefer et al., 2015; Coats et al., 2018), which in combination with variability in pore geometry, crystallinity and other textural parameters of volcanic rocks ensures that scatter

in volcanic rock strength is high (Lavallée and Kendrick, 2020; Heap et al., 2016b). This variability is exacerbated by the effects of pore pressure (Farquharson et al., 2016), in-situ temperature (Coats et al., 2018; Lamur et al., 2018), chemical alteration (Pola et al., 2014; Wyering et al., 2014; Farquharson et al., 2019), thermal stressing (Kendrick et al., 2013; Heap et al., 2014b) and time-dependent (Heap et al., 2011) or cyclic (Schaefer et al., 2015; Benson et al., 2012) stressing, whose impact is contrasting in different volcanic rocks, further enhancing the range of mechanical properties of materials that construct

volcanic edifices and lava domes.

During laboratory deformation, acoustic emissions (AEs) can be recorded; AEs are produced by the creation, propagation and coalescence of fractures which accelerates in the approach to failure, forming the basis for various forecasting approaches (e.g. Kilburn, 2003; Bell et al., 2011; Bell, 2018; Voight, 1988). The frequency-amplitude distribution of AEs are commonly

observed to follow an exponential distribution (e.g. Pollock, 1973; Scholz, 1968). This distribution is analogous to the Gutenberg-Richer relation observed for the frequency-magnitude distribution of tectonic earthquakes (Gutenberg and Richter, 1949). Accordingly, 'b-value' may be calculated for the distribution of AE amplitudes, describing the relative proportions of small and large events. Previous work on a broad range of lithologies showed that b-value is higher during ductile (compactant) deformation as cracking events are pervasively distributed, than during brittle (dilatant) deformation which is often localised

(Scholz, 1968). In their study on porous glasses Vasseur et al. (2015) showed that b-value increases as a function of heterogeneity (~ porosity) due to the number of nucleation sites in heterogenous materials that allow pervasive damage. Complementary work on three-phase magmas (glass, crystals and pores) showed that b-value depended on the applied stress, with higher stresses resulting in faster deformation, more localised damage zones and correspondingly lower b-values (Lavallée et al., 2008). Similarly, during a single episode of deformation, results on various rocks and glasses have also

indicated that b-value decreases as damage accrues and strain becomes localised to a damage zone or failure plane (Vasseur et



al., 2015; Lockner, 1993; Meredith et al., 1990; Main et al., 1992), whilst in double direct shear, smoother, less heterogenous fault surfaces produced lower *b*-values during slip (Sammonds and Ohnaka, 1998).

Elastic moduli also elucidate materials' response to deformation, and are measured from mechanical data or from ultrasonic
velocity measured in the laboratory (though values do not necessarily correlate; Kendrick et al., 2013; Heap et al., 2020); in particular Young's modulus indicates the stress-strain response to loading and primarily correlates negatively and poorly with porosity (Heap et al., 2020 and references therein). Ultransonic velocity is itself an indicator of material properties (e.g. Vanorio et al., 2002); for volcanic rocks and magmas both P- and S-wave velocity, and their ratio, depend on the mineralogical assemblage (Caricchi et al., 2008; Vanorio et al., 2002), porosity (vesicularity or fracture damage; e.g. Lavallée et al., 2013;
Lesage et al., 2018) and temperature (e.g. Scheu et al., 2006a). During deformation in compression, seismic velocity has been shown to first increase and then more substantially decrease as damage accrues (Ayling et al., 1995), which has been linked via AE monitoring to the generation of fractures (Benson et al., 2007; Zhang et al., 2019). Whilst seismic velocity is a valuable characterisation tool, it is sensitive to the degree of saturation (pertinent to wet volcanic systems) and difficult to measure during dynamic testing due to the generation of AEs (Zhang et al., 2019), as well as being both technologically and
computationally challenging (Benson et al., 2007). Coda wave interferometry (CWI) has been employed as an alternative, being sensitive to small fluctuations in material properties (e.g. Singh et al., 2019; Snieder et al., 2002; Griffiths et al., 2018), including crack damage (Lamb et al., 2017) or degree of saturation (Grêt et al., 2006). The utilisation of CWI at active volcanic systems has not only tracked migrating seismic sources (e.g. Lamb et al., 2015), but has also indicated velocity reduction prior to eruptions on an equivalent scale to that measured in the laboratory (Erdem and Waite, 2013; Lamb et al., 2017; Haney et
al., 2014), validating its implementation in rock physics to track material evolution.

The spectrum of lab-based approaches offer an idealised picture of material characteristics of a given volcanic system, representing intact-rock values of material coherent enough to sample. Utilisation of field-based measurements using the Schmidt hammer (Harnett et al., 2019a), or in-situ porosity and permeability measurements (Mordensky et al., 2018) have
been employed in combination with laboratory testing in an attempt to examine the representativeness of sample selection at volcanoes (e.g. Bernard et al., 2015; Schaefer et al., 2015). Thomas et al. (2004) used the rock-mass rating (RMR) index and the Hoek Brown criterion to deduce that edifice strengths likely show a 96 % reduction from intact rock strength measured in the laboratory due to rock mass discontinuities and surface conditions. Whilst it is not necessarily the responsibility of those conducting mechanical tests to apply such corrections, it is vital that such considerations are made in the modelling and
assessment of hazards posed by partial or complete collapse of volcanic edifices and lava domes.

### 1.2 Mount Unzen eruption and lavas

In order to understand how physical and mechanical properties of volcanic rocks vary we must first consider the variability from a single lava dome eruption at a volcanic system. The 1990 – 1995 eruption at Mount Unzen, on the Shimabara Peninsula



(Fig. 1a) began on 17[th] November 1990 and the extrusion of lava at the Gigoku-ato crater commenced on 20[th] May 1991 (Nakada and Fujii, 1993). A total of $1.2 \times 10^8\,m^3$ lava was erupted via endogenic and exogenic growth, with approximately half this volume preserved in the Heisei-Shinzan lava dome (Nakada et al., 1999). Endogenic versus exogenic growth has been modelled to be controlled by extrusion rate (Hale and Wadge, 2008): During May 1991 – November 1993 effusion rates were high (Nakada et al., 1995; Nakada and Motomura, 1999), resulting in the formation of 13 lava lobes (Sato et al., 1992; Nakada

and Fujii, 1993) by exogenous growth. Effusion rates waned after November 1993 and dome growth was endogenous until mid-October 1994 when a lava spine extruded in the centre of the dome surface (Saito and Shikawa, 2007; Nakada and Motomura, 1999). As the dome grew, new lobes were extruded into older collapse scars, which formed planes of weakness that facilitated further collapses (Nakada et al., 1999). Throughout the eruption numerous collapse events caused block-and-ash flows and rock falls (Sato et al., 1992), as the lava dome was constructed atop the steep substratum (Brantley and Scott,

1993). Tragically one such collapse on 3[rd] June 1991, led to the death of 43 people To date, the lava dome remains unstable (Shi et al., 2018), the frontal portion of lobe 11 continues to move SE-ESE at rate of 2.45-5.77 cm per year (over the last decade) presenting the risk of collapse of a portion of the lava dome up to $10^7\,m^3$ in size (Hirakawa et al., 2018). As such the summit and large proportion of the flanks remain an exclusion zone.

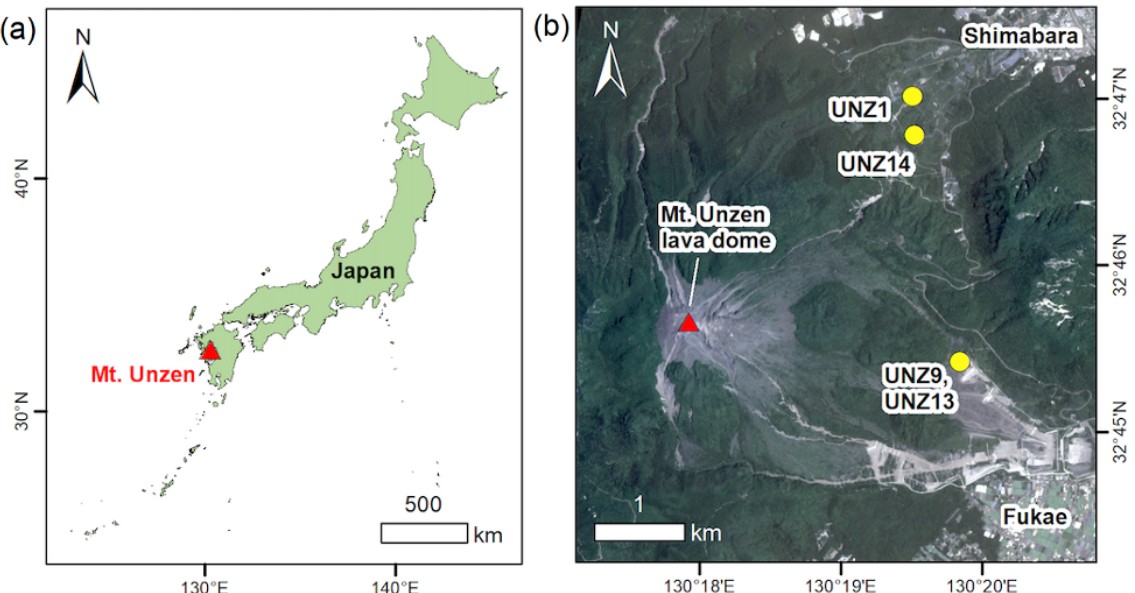

**Fig. 1: (a) Location of Mt. Unzen on the Nagasaki peninsula of Japan (country boundary data provided by the World Resource Institute). (b) Location of samples used in the study shown on top of a multispectral PlanetScope Scene with 3 m spatial resolution from 7th June 2020 (Planet, 2017).**



The erupted lavas are porphyritic dacites with abundant, large phenocrysts and significant porosity (typically > 10 %), which
       is distributed heterogeneously (Nakada and Motomura, 1999; Noguchi et al., 2008; Cordonnier et al., 2009; Bernard et al.,
       2015; Coats et al., 2018; Scheu et al., 2008). Much of the lava exhibits anisotropic textures, and shear zones pierce the lava
       dome carapace, relics of shallow conduit strain localisation in the hot, viscous magmas (Wallace et al., 2019; Miwa et al.,
       2013; Hornby et al., 2015). Ongoing fumarole activity and prolonged residence at elevated temperature has resulted in
substantial hydrothermal alteration in localised areas of the dome (e.g. Almberg et al., 2008). Numerous experimental
       investigations have examined the porosity distribution, rheology, strength, seismic velocities, elastic moduli, fragmentation
       threshold and frictional coefficients of the 1991-1995 lavas (Scheu et al., 2006a; Scheu et al., 2008; Cordonnier et al., 2009;
       Kremers et al., 2010; Hornby et al., 2015; Bernard et al., 2015; Coats et al., 2018; Lavallée et al., 2007; Kueppers et al., 2005),
       painting a picture of a highly heterogeneous lava dome. Understanding this heterogeneity in terms of physical and mechanical
variability is vital. Coats et al. (2018) showed that these dacitic rocks weaken as they cool from magmatic to ambient
       temperatures, and as the deformation of rocks is inherently time-dependent (Dusseault and Fordham, 1993) the hazards at
       Mount Unzen continue to evolve, especially in light of the potential for regional earthquakes or renewed volcanic unrest.

       Here, by utilising the range of materials produced during a single eruption at Mount Unzen, we demonstrate the importance of
material characterisation. Mount Unzen represents an ideal case study as the eruptive products exhibit mostly invariable
       chemical and mineralogical attributes, and they have experienced similar eruptive and cooling history; thus their study allows
       a robust description of relationships between physical and mechanical characteristics. We assess the contribution of rock
       porosity and anisotropy on rock strength under dry and water-saturated conditions, and examine Young's modulus, as well as
       the interrelation of these properties and rock permeability. We assess the temporal evolution of damage during laboratory
compressive and tensile deformation using acoustic monitoring of crack damage, examining accelerating rates of energy
       release and tracking progression of seismic *b*-value. We also employ coda wave interferometry during deformation to further
       quantify progression of damage during stressing. Such investigations that consider damage progression and strength as a
       function of porosity, anisotropy and saturation under different deformation modes are important in our interpretation of volcano
       monitoring data, elucidating the processes responsible for observed characteristics and defining their associated hazards.


**2 Materials and Methods**

**2.1 Sample selection and characterisation**

**2.1.1 Sample collection**

Unzen lavas are typically porphyritic dacites with ~ 63 wt. % $SiO_2$, rhyolitic interstitial glass. Lavas from the collapse deposits
of the 1991-95 lava dome sampled in this study have been described as having variable porosities of approximately 10-35 %
       (Kueppers et al., 2005; Coats et al., 2018; Hornby et al., 2015) and crystallinity (including microlites) of up to ~ 75 %, including
       large (> 3 mm) and abundant (> 25 vol %) plagioclase phenocrysts, along with fewer amphibole (~ 5 vol %), biotite (~ 2 vol
       %) and quartz (~ 2 vol %) phenocrysts and microphenocrysts set in a partially crystalline (30-55 vol %) groundmass of





plagioclase, pyroxene, quartz, pargasite and iron-titanium oxides (Coats et al. 2018; Nakada and Motomura, 1999; Wallace et
al., 2019).

A suite of blocks, each > 15 kg were collected from block-and-ash flow deposits on the eastern and north-eastern flanks during
a field campaign in 2015, and 4 blocks representative of the porosity and textures observed in the erupted lavas (cf. Kueppers
et al. 2005) were chosen for this study (Fig. 1b). Broadly the sample blocks chosen span low (UNZ14), medium (UNZ1) and
high (UNZ13) porosity for the range observed in the erupted lavas, plus an additional block that displays an anisotropic
cataclastic fabric (UNZ9). Blocks UNZ1 and UNZ13 were also used for the study by Coats et al. (2018) which examined the
rheological response to deformation at high temperature and defined a failure criterion for porous dome rocks and lavas.

**2.1.2 Sample preparation**
Samples were cored using a pillar drill at University of Liverpool to prepare cylinders of both 20 mm and 40 mm diameter. In
the case of the anisotropic block, cores were prepared both parallel (UNZ9a) and perpendicular (UNZ9b) to the plane of the
fabric, producing five sample groups: UNZ1, UNZ9a, UNZ9b, UNZ13 and UNZ14 (Fig. 2). The 20 mm cylinders were cut
and ground plane-parallel to a nominal length of 40 mm to prepare samples (herewith termed cores) for porosity, unconfined
gas permeability, confined water permeability and both dry and water saturated uniaxial compressive strength (UCS)
measurements. The 40 mm cylinders were cut to lengths of 20 mm to prepare samples (termed discs) for porosity and
unconfined gas permeability measurements and indirect tensile strength testing (UTS) of dry and saturated samples using the
Brazilian disc method (Fig. 2). Depending on material availability, between 7 and 14 each of both cores and discs were
produced for each of the five sample groups resulting in a total of 114 cores and disks. In addition, sample offcuts of each rock
were ground to a fine powder for solid density measurements (see below).

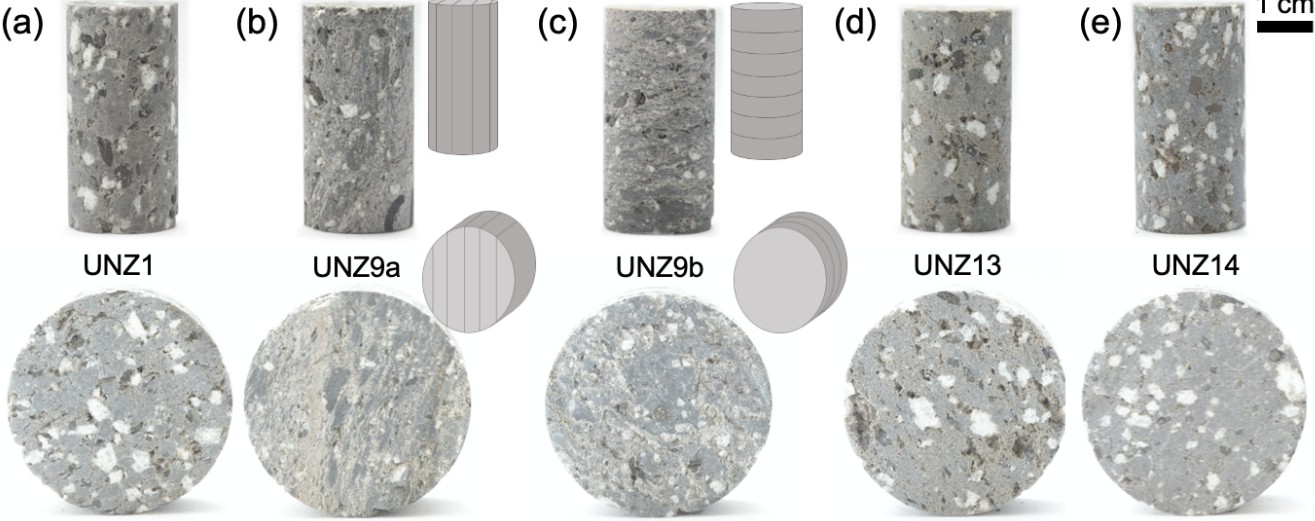



**Fig. 2: Photographs of samples used for uniaxial compressive strength (UCS) tests and discs used for indirect tensile strength testing (UTS) using the Brazilian disc method: (a) UNZ1, an isotropic dacite of medium porosity with large euhedral phenocrysts and pore space comprising vesicles adjacent to phenocrysts and microfractures traversing the groundmass; (b) UNZ9a, a cataclastic dacite with clear sub-parallel cataclastic**
**banding comprised of fractured phenocrysts and porous fault gouge cored parallel to the fabric (see inset schematic), both void space and crystals (often fragments) are smaller than in isotropic sample UNZ1; (c) UNZ9b, the same cataclastic sample cored perpendicular to the fabric (see inset schematic); (d) UNZ13, the lowest density sample with large phenocrysts, large sub-rounded vesicles, and varying degrees of coalescence often positioned in close proximity to crystals; (e) UNZ14, the densest sample selected for the study also has**
**large euhedral phenocrysts, with typically smaller pores and fine fractures traversing the dense groundmass. For all samples, direction of the principal applied stress during testing is vertical.**

### 2.1.3. Microstructural characterisation

Thin sections were prepared with fluorescent dyed epoxy from the offcuts of sample cores in the same orientation as coring
direction. Thin sections were imaged using a DM2500P Leica microscope with both reflected light with UV filter to examine microstructures and in plane polarised transmitted light to assess mineralogy.

### 2.1.4 Porosity determination

The porosity of all 114 cores and discs was assessed using an AccuPyc 1340 helium pycnometer from Micromeritics using a
35 cm$^3$ cell (to provide volume with an accuracy of ±0.1 %). Sample length ($l$; in centimetres), radius ($r$; in centimetres) and mass ($m$; in grams) were recorded, providing sample density ($\rho_s$; in grams per cubic centimetre) via:

$$\rho_s = \frac{m}{\pi r^2 l} \qquad (1)$$

The solid density of the rocks ($\rho_0$) was determined in the pycnometer by measuring the volume of ~ 25 g aliquots of the
powders from each sample block, and total porosity ($\emptyset_T$) was calculated via:

$$\emptyset_T = 1 - \frac{\rho_s}{\rho_0} \qquad (2)$$

To constrain the fraction of isolated pores in the rocks, the material volume was measured ($V_m$; in cubic centimetres) for each core and disc sample in the pycnometer. The connected porosity ($\emptyset_c$) of the samples was then determined via:

$$\emptyset_c = 1 - \frac{V_m}{\pi r^2 l} \qquad (3)$$

and isolated porosity ($\emptyset_i$) via:





$$\emptyset_i = \emptyset_T - \emptyset_c \tag{4}$$

The connected porosity is the most robustly defined parameter (as total and isolated porosity rely on powder density determined for the bulk sample, not the specific core). As such, connected porosity was further analysed for average porosity, standard deviation and coefficient of variation for each sample group (UNZ1, UNZ9a, UNZ9b, UNZ13 and UNZ14) and was used for presenting all permeability measurements and mechanical results framed in terms of porosity.

**2.1.5 Unconfined gas permeability**

Permeability of the cores and discs at ambient atmospheric conditions was measured using a TinyPerm II minipermeameter from New England Research Inc. The apparatus utilises the pulse decay method via an 8 mm circular aperture in contact with the sample surface, thus sampling a different volume depending on absolute permeability (Filomena et al., 2014). This method provides permeability determination with an accuracy of ~ 0.2 log units at low porosities to 0.5–1 log units at higher porosities

(Lamur et al., 2017). For each of the samples 5 measurements were made at different positions on each flat face of the rock sample (10 total per sample). In total, 1065 unconfined gas permeability measurements were made on the samples, and the values were used to determine the average, standard deviation and coefficient of variation for each sample and additionally for each sample group (UNZ1, UNZ9a, UNZ9b, UNZ13 and UNZ14).

Additionally, for 2 of the blocks, the macroscopically anisotropic UNZ9 and the densest block UNZ14, the block had to be cut in order to prepare the sample cores, revealing planar surfaces of up to 8 x 40 and 8 x 18 cm$^2$ respectively. The planar surfaces of the dissected blocks were additionally mapped using the TinyPerm II minipermeameter at a grid spacing of 1cm. An additional 262 measurements were made for sample UNZ9 and 117 for sample UNZ14. The values were used to determine the average, standard deviation and coefficient of variation for each sample group.


**2.1.6 Confined water permeability**

A subset of 3 cores from each of the five sample groups (UNZ1, UNZ9a, UNZ9b, UNZ13 and UNZ14) were chosen to determine permeability as a function of confining pressure. Using a hydrostatic pressure cell developed by Sanchez Technologies the permeability of the samples was measured using the steady-state flow method. Confining pressure ($P_c$) was

set to increments of 5.5, 9.5 and 13.5 MPa and at each increment flow rate ($Q$) was varied until steady state flow was achieved. Pore pressure differential ($\Delta P$) was calculated by monitoring pressure upflow and downflow from the sample (held between 1.1-1.5 MPa) during steady state flow and the average subtracted from the confining pressure to define the effective pressure ($P_{eff}$) for the measurements. Permeability ($k$) was determined at each $P_{eff}$ via Darcy's law:

$$k = \frac{Q\mu l}{a\Delta P} \tag{5}$$






where $\mu$ is the water viscosity, $l$ is the sample length and $a$ is the sample cross-sectional area. Thus, the effect of increasing effective pressure on permeability and the sensitivity to confinement (cf. burial) of each sample was revealed.

## 2.2 Sample deformation

### 310   2.2.1 Uniaxial compressive testing

From each suite of samples, 4 cores were selected at random for mechanical testing, including one core which had been measured for water permeability (section 2.1.6). Uniaxial compressive strength (UCS) tests were performed on three dry cores and one saturated core from each sample group using a 100 kN Instron 8862 uniaxial press with FastTrack 8800 tower and Instron Dynacell 2527 load cell in the Experimental Volcanology and Geothermal Research Laboratory at University of 315   Liverpool. Two ceramic piezoelectric transducers (PZT) were attached on the samples during testing (described in section 2.2.4). A constant compressive strain rate of $10^{-5}$ s$^{-1}$ was used for testing, with load and axial displacement recorded at a rate of 100 Hz. The Bluehill® 3 software was used to compute compressive stress and strain ($\varepsilon$) during deformation using sample dimensions. The end of each experiment was defined by a stress drop exceeding 20 % of the monitored normal stress. All mechanical data were corrected for the compliance of the set-up at the relevant experimental deformation rate. Following Coats 320   et al., (2018) Young's modulus was calculated from the linear elastic portion of the stress strain curve picked using an automated script written in MATLAB (Coats, 2018).

### 2.2.2 Brazilian disc testing

From each suite of samples, 4 discs were selected at random for mechanical testing. The Brazilian disc method to determine 325   indirect tensile strength (UTS) was performed on three dry discs and one saturated disc from each sample group using the same 100 kN Instron 8862 uniaxial press with FastTrack 8800 tower and Instron Dynacell 2527 load cell. Two ceramic piezoelectric transducers (PZT) were attached on the samples during testing (described in section 2.2.4). In these tests the disc shaped specimens were loaded diametrically on flat loading platens at an equivalent diametric strain rate of $10^{-5}$ s$^{-1}$. Methods and standards utilised for Brazilian disc testing are frequently conflicting in terms of deformation/ loading rate and time to 330   failure (ISRM, 1978; ASTM, 2008; Li and Wong, 2013; Hornby et al., 2019), here we adopt the approach of Lamb et al. (2017). All mechanical data were corrected for the compliance of the set-up at the relevant experimental deformation rate. The Bluehill® 3 software was used to monitor axial displacement and load (N) at 100 Hz, and the conversion to tensile stress ($\sigma_t$) was made in real time via:

$$\sigma_t = \frac{2N}{\pi dl} \tag{6}$$


where $d$ is diameter and $l$ is thickness (length) of the disc (ISRM, 1978). The end of each experiment was defined by a stress drop exceeding 20 % of the calculated stress.



### 2.2.3 Interrelation of mechanical properties

Since the compressive and tensile strength and Young's modulus of rocks all show a dependence on porosity (as has been well documented in the literature; e.g. Lavallée and Kendrick, 2020 and references therein; Heap et al., 2020) we define the interrelation of these parameters to provide useful first-order constraints of material properties as a function of porosity. We do so by employing least squares regressions to ascribe power law relationships to compressive strength, tensile strength and Young's modulus as a function of porosity of the eruptive products. We then combine these equations to define the interrelation

of each parameter, and to express their evolving relationships as a function of porosity. We limit our analysis to the porosity range examined here (between the 1st-99th percentile), and add the caveat that these relationships are likely to be lithologically-dependent due to the textural and microstructural nature of materials (Lavallée and Kendrick, 2020), yet are likely to be broadly applicable to glassy, porphyritic volcanic rocks.

### 2.2.4 Acoustic emissions - passive

Two ceramic piezoelectric transducers (PZT) were attached on the samples during both UCS and UTS tests. In the UCS set-up PZTs were housed within specially machined spring-loaded platens that allowed direct contact at the ends of the sample cores, whilst in the UTS set-up transducers were placed on diametrically opposing edges of the Brazil discs, perpendicular to the direction of axial loading (See Fig. S1). The sensors monitored acoustic emissions (AEs) released during deformation at a

sampling rate of 1 MHz. These signals were first fed through 20 dB amplifiers before reaching a PAC PCI-2 two-channel recording system with a bandwidth of 0.001-3 MHz, allowing hit-based collection and waveform streaming. For each experiment The timing and energy of each event were recorded, and an amplitude cut-off of -3.3 was chosen. the timing of each event was recorded, AEs generated by pulsing were excluded (see section 2.2.5) and the energy of each hit was calculated using the root-mean-square of the recorded waveform following the method of Lamb et al. (2017).


The acceleration of acoustic emission rate was assessed by fitting Poisson point process models to the first 75 % of the event time series after this point, the quick succession of events hinders distinction and can lead to artificial reduction of event rate) for each experiment (excluding events below an amplitude of -3.3). The model assumed an exponential acceleration (after Voight, 1989) in the rate of acoustic emissions with time:

$$\frac{d\Omega}{dt} = ke^{\lambda(t-t_0)} \tag{7}$$

where the parameter $k$ relates to the absolute amplitude of the acceleration, whilst $\lambda$ is the exponential rate parameter. For this analysis only dry tests were used, since it proved impossible to distinguish passive and active AE events for saturated samples. Models are fitted using a Bayesian MCMC method (Ignatieva et al., 2018; Bell et al., 2018; Bell, 2018), and model parameters

($k$ and $\lambda$) reported as the maximum *a posteriori* values. The parameter $k$ relates to the absolute amplitude of the acceleration, whilst $\lambda$ is the exponential rate parameter. The frequency amplitude distribution of the AEs from each test were plotted and





from this the *b*-value for each experiment was calculated using the maximum-likelihood method (after Roberts et al., 2015). In addition, the *b*-value was determined for each third of the test to examine evolution (*Δb*) during deformation.

### 2.2.5 Acoustic emissions - active

In addition to passive monitoring of acoustic emissions, to calculate elastic velocity properties of the samples, one PZT was set to produce "pulses" for the entire experiment duration while the other PZT recorded the pulses after they travelled through the sample. The pulses were released in "bursts" of five events, spaced 0.5 s apart, and triggered every 5 s. Following the method of Lamb et al. (2017) the received bursts were stacked to increase the signal-to-noise ratio, and coda wave interferometry (CWI) was applied to the stacks. This method utilises the degree of correlation between stacked waveforms at different time intervals, compared to the reference (here the first stacked pulse) to calculate the variance of the travel time perturbation, and thus to calculate relative change in velocity during the experiment (for further details of the method see Lamb et al. 2017).

### 3 Results

### 3.1 Textures, microstructures and mineralogy

The dacitic samples were deposited by block-and-ash flows during growth and collapse of the lava dome during the 1990-1995 Heisei eruption of Mount Unzen (e.g. Sato et al., 1992). The lavas are porphyritic and partially glassy, and show variability in crystallinity, textures and microstructures (Fig. 2, Fig. 3). The porous networks are comprised of connected cracks and vesicles frequently concentrated around phenocrysts (Fig. 2, Fig. 3a-e). Despite local heterogeneities the pore network is relatively isotropic in samples UNZ1, UNZ13 and UNZ14 (Fig. 2, Fig. 3a, b & e, respectively), whereas the sample block selected due to the presence of cataclastic banding (UNZ9) observable in hand specimen (Fig. 2) shows strongly anisotropic pore structures (Fig. 3c-d). Texturally UNZ1 and UNZ13 are similar (Fig. 3); both samples show pores up to a few mm's in size either adjacent to or completely bounding crystals, and the groundmass hosts sub-rounded vesicles which are slightly more abundant in UNZ13, leading to the UNZ1 groundmass appearing denser. The UNZ1 groundmass hosts occasional narrow fractures (0 to 10's microns) that traverse the dense areas, extending up to 5 mm and connecting phenocrysts (Fig. 3). Sample UNZ14 has notably fewer vesicles, and again, fine cracks (here finer than in UNZ1, typically < 10 microns) that are more abundant and of greater length-scale (occasionally > 10 mm) than in UNZ1, which traverse dense areas of groundmass, and pass along crystal margins (Fig. 2, Fig. 3). In block UNZ9 the cataclastic fabric was cored in two orientations to produce sample UNZ9a parallel to the fabric and UNZ9b perpendicular to the fabric (Fig. 2, Fig. S1). The thin sections represent a core of each cut vertically (UNZ9a in Fig. 3b & g, UNZ9b in Fig. 3c & h,) to highlight the fabric with respect to compression direction in later strength tests (note that the brazil discs are diametrically compressed). The UNZ9 samples comprise variably porous cataclastic bands with fragmental phenocrysts (Fig. 2, Fig. 3). Porosity is thus anisotropically distributed across denser and more porous bands, though still typically focused around crystals, here often crystal fragments, and is similar in abundance to the porosity of UNZ1.

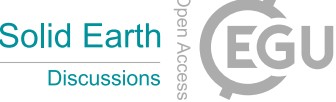

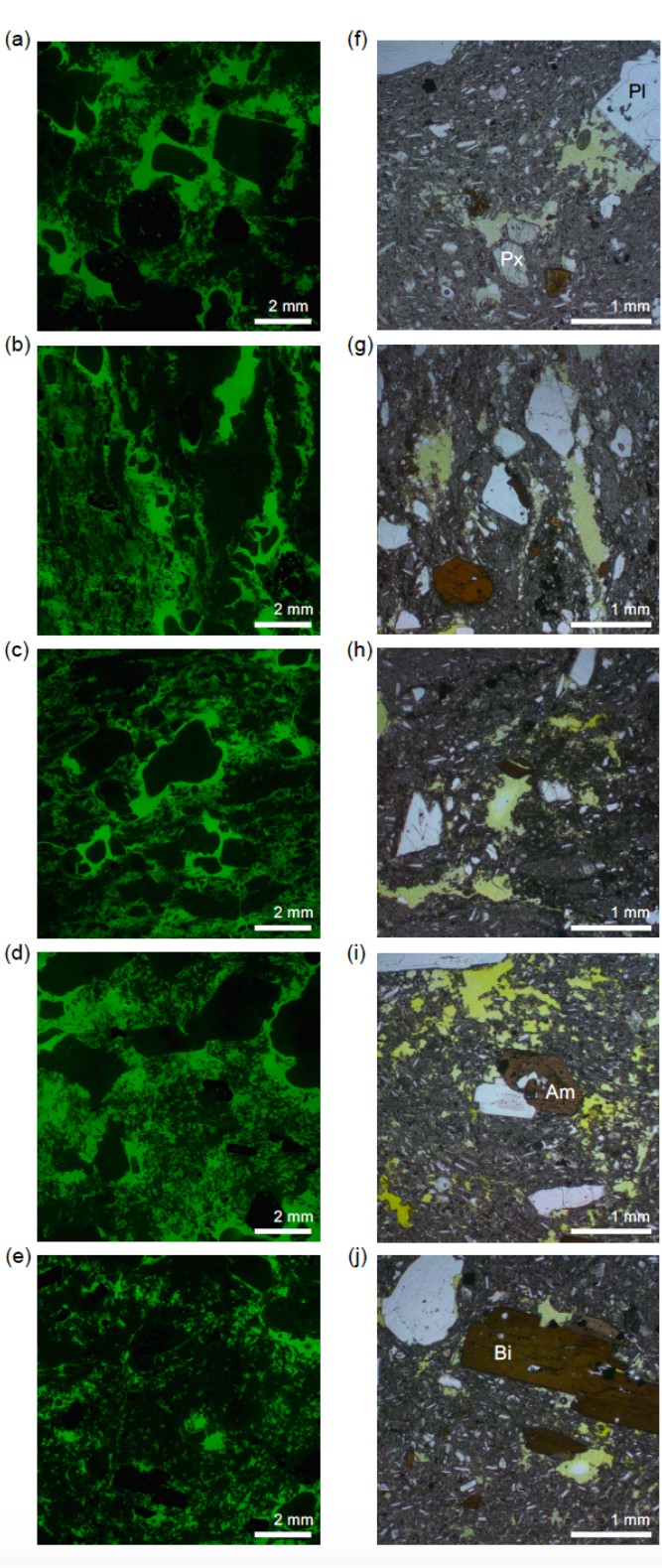



**Fig. 3: Images of thin sections in reflected light with a UV filter (a-e) and plane polarised light (f-j) showing the range of textures of the tested materials: (a) UNZ1 has pore space concentrated around phenocrysts, occasional sub-rounded vesicles and a relatively dense groundmass that hosts occasional fractures of 0-20 microns width, up to ~ 2 mm in length; (b) UNZ9a hosts pore space concentrated in laterally extensive bands in the orientation of cataclastic fabric observable in hand specimen (vertical), which are interspersed by denser bands, pores border angular fragmental crystals; (c) UNZ9b shows the same textures as in UNZ9a, here orientated horizontally, and fine fractures are additionally visible within the groundmass and broken phenocrysts (note the large rounded black patch in the centre is a poorly-impregnated pore, not a sub-rounded phenocryst); (d) UNZ13 has distinguishing sub-rounded vesicles in the groundmass and concentrated around phenocrysts, phenocrysts occasionally host a number of very fine fractures (note the large rounded black patch at the top right is a poorly-impregnated pore); (e) UNZ14 shows smaller pores more distributed, but still localised adjacent to phenocryst, occasional thin (< 10 micron) fractures of up to ~ 6 mm propagate through the groundmass connecting phenocrysts which themselves are highly fractured (with hairline fractures). Transmitted light images (f-j) allow the identification of plagioclase (Pl, > 25 vol. %), amphibole (Am, ~ 5 vol. %), biotite (~ 5 vol. %) and pyroxene (Px, < 2 vol. %) phenocrysts and microphenocrysts (quartz is present but not shown) and glassy groundmass with microlites of plagioclase, pyroxene, quartz, amphibole and iron-titanium oxides of 10-100 microns (~ 30 vol. %). Plagioclases show occasional zoning (i) and concentric bubble trails (f and j). The cataclastic samples UNZ9a (g) and UNZ9b (h) additionally have broken crystals, most frequently plagioclase that form trails parallel to elongate porosity-rich bands, and the groundmass shows heterogeneously distributed light and dark zones.**

The samples have large (often > 3 mm) phenocrysts (Fig. 3f-j) which are easily identifiable in hand specimen (Fig. 2), the largest and most abundant of which are plagioclase (> 25 vol. %), followed by amphiboles (~ 5 vol. %) and frequently fractured biotites (~ 5 vol. %) with smaller and more infrequent quartz and pyroxenes (each < 2 vol. %), with the same minerals also forming microphenocrysts (Fig. 3), as has previously been described for Unzen lavas (e.g. Nakada and Motomura, 1999). The glassy groundmass hosts microlites of 10-100 microns of plagioclase, pyroxene, quartz, pargasite and iron-titanium oxides that make up approximately 30 vol. %, in keeping with previous observations of groundmass crystallinity, which slowly increased from ~ 30 to ~ 50 vol. % throughout the eruption (Nakada et al., 1995; Nakada and Motomura, 1999). The cataclastic bands of sample UNZ9 host angular fragments of crystals, some of which are retained in fragmental lenses of single minerals (Fig. 3g & h), the relics of grain size reduction compared to the pristine lavas of UNZ1, UNZ13 and UNZ14 (Fig. 2, Fig. 3), as has been noted in other conduit fault zone products at Mount Unzen (e.g. Wallace et al., 2019).

### 3.2 Porosity and porosity variability

—





Across the suite of 114 samples, total porosity determined by helium pycnometry ranged from 9.14 to 42.81 %, with a significant range observed within each sample group (see Table 1; Table S1). The average total porosity for each sample group spanned a narrower range of 16.05 to 36.46 %, ranking the samples as follows from least to most porous: UNZ9a, UNZ14, UNZ9b, UNZ1, UNZ13 (Table 1). Density ranged from 1.54-2.40 g.cm$^{-3}$ (Table S1) closely matching previously constrained densities of the eruptive products of 1.6-2.4 g.cm$^{-3}$ with bimodal distribution (Kueppers et al., 2005). The solid density of the

5 sample types spanned a narrow range of 2.64-2.67 g.cm$^{-3}$, representing the similarity in constituent phases of the lavas. The degree of isolated porosity ranged from 0.39 to 5.37 %, and was variable within a single sample group, typically with a minor increase with increasing total porosity (Fig. 4a, Table 1, Table S1) as has been previously observed for the eruptive products at Mount Unzen (Coats et al., 2018). Notably, the anisotropic samples (UNZ9a and UNZ9b) had higher connectivity (lower isolated porosity; Table 1) than isotropic samples with similar porosity (fall closer to 1:1; Fig. 4a). Connected porosity of the

114 samples ranged from 7.47 to 40.12 %, and averages of each of the 5 sample groups ranged from 13.69 to 33.13 %, ranking the samples by connected porosity (different to the ranking in total porosity) as follows from least to most porous: UNZ14, UNZ9a, UNZ9b, UNZ1, UNZ13 (Table 1). The standard deviation within a single sample group was generally higher for higher porosity. Variability within each sample group can be better evaluated by considering the coefficient of variation; the isotropic samples (from least to most porous; UNZ14, UNZ1 and UNZ13) have lower coefficients of variation (6.62, 6.23, and

9.97 %, respectively) whilst the anisotropic samples (UNZ9a and UNZ9b) have higher coefficients of variation (23.24 and 16.25 % respectively). Porosity and density constrained here closely match, and span the range of, lavas previously measured for the 1990-95 dome eruption (Coats et al., 2018; Cordonnier et al., 2008; Hornby et al., 2015; Kueppers et al., 2005; Wallace et al., 2019).

Table 1: Sample (core and disc) porosity and unconfined permeability overview.

| Sample | Number of samples | Average total porosity | Average isolated porosity | Connected porosity | | | Unconfined permeability | | |
|---|---|---|---|---|---|---|---|---|---|
| | | | | Average | Standard deviation | Coefficient of variation | Average | Standard Deviation | Coefficient of variation |
| | | % | % | % | | % | m² | | % |
| UNZ1 | 24 | 21.33 | 2.70 | 18.64 | 1.16 | 6.23 | 3.05E-12 | 3.30E-12 | 108.38 |
| UNZ9a | 20 | 16.05 | 1.31 | 14.73 | 3.42 | 23.24 | 1.89E-13 | 1.65E-13 | 87.24 |
| UNZ9b | 19 | 18.86 | 1.64 | 17.22 | 2.80 | 16.25 | 2.19E-13 | 1.44E-13 | 65.98 |
| UNZ13 | 27 | 36.46 | 3.32 | 33.13 | 3.30 | 9.97 | 2.89E-11 | 5.51E-11 | 190.66 |
| UNZ14 | 24 | 16.08 | 2.39 | 13.69 | 0.91 | 6.62 | 1.93E-13 | 1.82E-13 | 94.15 |

## 3.3 Permeability

### 3.3.1 Unconfined permeability and permeability variability

The permeability of the cores and discs was measured at ambient atmospheric conditions using up to 10 measurements on

different parts of the sample surface to assess local variations in permeability. The range of all 1065 measurements spanned



1.54 x10$^{-14}$ to 2.67 x10$^{-10}$ m$^2$ (Fig. 4b) with standard deviations of permeability of up to 6.01 x10$^{-10}$ m$^2$ within a single core or disc, corresponding to a coefficient of variation of over 259 % (see Table S1 [N.B where coefficient of variation was less than 10 % after 5 measurements, no further measurements were made]). The permeability shows a positive correlation with porosity; Fig. 4b shows the 1065 individual measurements made on 114 samples as well as the averages for each core or disc used for

further testing. The average permeability may span > 2 orders of magnitude for a given porosity, despite the large scatter of permeability for an individual core the distinct grouping of the sample suites (i.e. UNZ1, UNZ9a, UNZ9b, UNZ13, UNZ14) is clearly observable. Notably, permeability is lower for the macroscopically anisotropic sample UNZ9b than for the macroscopically isotropic sample UNZ1 of similar porosity, though no such discrepancy is noticed with macroscopically isotropic UNZ14 (large symbols in Fig. 4b, Table S1).


We additionally used the permeability of each core and disc to collate the average permeability, standard deviation and coefficient of variation of each sample group (Table 1). Interestingly the permeability of the anisotropic samples cut parallel (UNZ9a) and perpendicular (UNZ9b) converge to similar averages despite plotting somewhat distinctly in porosity-permeability space (Fig. 4b). The standard deviation and coefficient of variability of permeability are notably higher for the

most porous, permeable sample UNZ13 (Table 1), which also has the largest absolute range in connected porosity of more than 15 % (Fig. 4b, Table S1). The anisotropic samples have the lowest coefficients of variation of permeability, despite having the largest coefficient of variation of porosity (Table 1).

Table 2: Planar block surface unconfined permeability measurement overview.

| Sample block | Surface covered | Number of measurements | Permeability | | | | |
|---|---|---|---|---|---|---|---|
| | | | Minimum | Maximum | Average | Standard deviation | Coefficient of variation |
| | *cm²* | | *m²* | | | | *%* |
| UNZ9 | 8 x 40 | 262 | 1.90E-15 | 1.51E-12 | 1.53E-13 | 2.19E-13 | 143.01 |
| UNZ14 | 8 x 18 | 117 | 9.15E-15 | 2.58E-12 | 1.75E-13 | 2.85E-13 | 163.07 |


As a final measure of permeability variation within the sample groups we additionally performed permeability measurements across the planar surfaces of the dissected sample blocks UNZ9 and UNZ14. The macroscopically anisotropic block UNZ9 was cut perpendicular to the direction of the cataclastic fabric, thus is geometrically equivalent to the sample group UNZ9a, whereas UNZ14 is macroscopically isotropic. Despite their textural differences the average porosity-permeability of the two

sample groups (determined on the cores) described above is very similar (1.89 x10$^{-13}$ m$^2$ and 1.93 x10$^{-13}$ m$^2$; Fig 3a and b). An additional 262 measurements were made for sample UNZ9 and 117 for sample UNZ14 (Table 2, Fig. S2). The averages for UNZ9 and UNZ14 were 1.53 x10$^{-13}$ m$^2$ and 1.75 x10$^{-13}$ m$^2$ respectively, very similar to those measured on the cores and discs. The permeability of UNZ9 had a slightly broader range, spanning almost 3 orders of magnitude, though the higher number of





measurements ensures similar standard deviation and coefficient of variation for each suite, which are notably significantly
       higher than across the cores and discs (Table 1; Table 2).

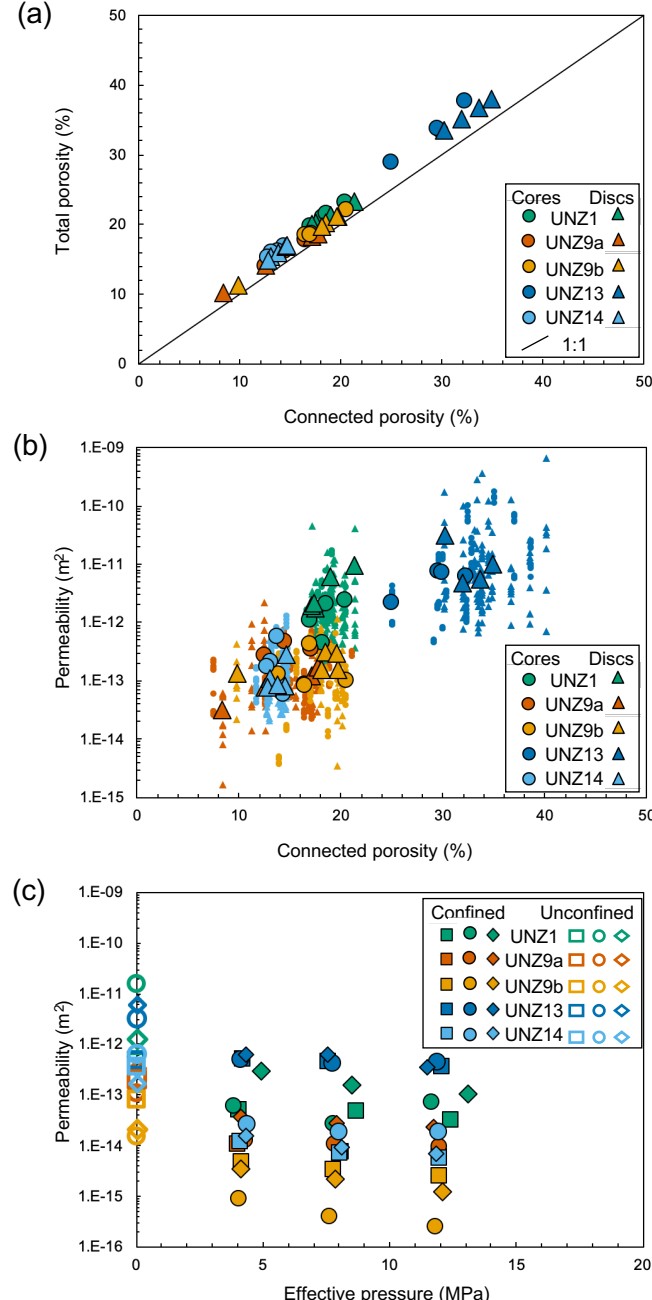

**Fig. 4: Physical attributes of tested dacite samples: (a) total versus connected porosity for the core and disc samples chosen for mechanical testing (complete dataset available in Table S1) with 1:1 marked highlighting**



the degree of connectivity; (b) Unconfined permeability as a function of connected porosity showing 1065
individual measurements of cores and discs (small symbols; see Table S1) measured using a gas
minipermeameter overlain by the average permeability of the samples selected for mechanical testing (large
symbols); (c) Unconfined and confined permeability as a function of effective pressure for 3 cores of each
material, unconfined measurements correspond to those in (b) and confined measurements are measured using
water in a pressure vessel (also plotted as a function of porosity in Fig. S3).

### 3.3.2 Permeability as a function of effective pressure

Permeability was measured for 3 samples from each group at 3 increments of confining pressure, and hence effective pressure
(Fig. 4c, Table 3). The permeability at the first increment of ~ 4 MPa is 1 to 2 orders of magnitude lower than the gas
permeability measurements made at atmospheric pressure conditions for all samples (section 3.3.1) and further decreases with
increasing effective pressure (Fig. 4c, Table 3). Here, the lowest permeability samples, with porosity comprised primarily of
microfractures (Fig 1), are most sensitive to effective pressure, with the largest reductions in permeability (Fig. 4c). Similarly
to the unconfined measurements, the most permeable samples at each effective pressure are isotropic UNZ13, followed by
isotropic UNZ1, isotropic samples UNZ14 are again very similar to UNZ9a, and the least permeable samples are UNZ9b.
Thus the cores cut parallel to the cataclastic fabric (UNZ9a) are significantly more permeable than those cut perpendicular
(UNZ9b), with a difference of more than an order of magnitude which was not noted in the permeability values of unconfined
measurements (Table 1).

Table 3:Permeability under confined conditions.

| Sample | Identifier | Porosity | Step 1 | | Step 2 | | Step 3 | |
|---|---|---|---|---|---|---|---|---|
| | | | Effective pressure | Permeability | Effective pressure | Permeability | Effective pressure | Permeability |
| | | % | MPa | m² | MPa | m² | MPa | m² |
| UNZ1 | 4 | 18.23 | 4.058 | 5.03E-14 | 8.703 | 4.80E-14 | 12.415 | 3.25E-14 |
| | 7 | 19.12 | 3.864 | 5.88E-14 | 7.768 | 2.76E-14 | 11.680 | 6.99E-14 |
| | 10 | 17.74 | 4.908 | 3.02E-13 | 8.470 | 1.59E-13 | 13.074 | 1.10E-13 |
| UNZ9a | 1 | 13.21 | 4.007 | 1.05E-14 | 8.101 | 7.58E-15 | 11.941 | 6.02E-15 |
| | 4 | 15.08 | 4.305 | 1.24E-14 | 7.825 | 1.04E-14 | 11.983 | 9.38E-15 |
| | 11 | 17.16 | 4.115 | 3.88E-14 | 7.900 | 2.90E-14 | 11.688 | 2.39E-14 |
| UNZ9b | 1 | 16.49 | 4.130 | 4.68E-15 | 7.789 | 3.41E-15 | 11.967 | 2.57E-15 |
| | 2 | 14.73 | 4.057 | 9.10E-16 | 7.630 | 4.01E-16 | 11.834 | 2.56E-16 |
| | 8 | 18.73 | 4.119 | 3.65E-15 | 7.808 | 2.30E-15 | 12.040 | 1.24E-15 |
| UNZ13 | 4 | 29.12 | 4.224 | 5.17E-13 | 7.595 | 4.68E-13 | 12.060 | 3.66E-13 |
| | 6 | 30.37 | 4.084 | 5.24E-13 | 7.734 | 4.31E-13 | 11.868 | 4.68E-13 |
| | 13 | 32.32 | 4.317 | 6.62E-13 | 7.553 | 6.62E-13 | 11.467 | 3.60E-13 |





| | | | | | | | |
|---|---|---|---|---|---|---|---|
| | 1 | 13.18 | 4.102 | 1.22E-14 | 8.031 | 7.29E-15 | 11.954 | 5.80E-15 |
| UNZ14 | 5 | 14.15 | 4.379 | 2.76E-14 | 7.957 | 1.95E-14 | 11.920 | 1.90E-14 |
| | 11 | 12.74 | 4.321 | 1.56E-14 | 8.062 | 9.70E-15 | 11.791 | 7.18E-15 |

## 3.4 Mechanical data

### 3.4.1 Strength in dry and saturated conditions

Stress-strain curves for all Uniaxial compressive strength (UCS) tests and Brazilian disc indirect tensile strength (UTS) data are shown in Fig. 5. The UCS of dry samples ranged from 13.48 to 47.80 MPa and was dominantly controlled by porosity (Fig. 6a), as has been previously observed for Mount Unzen lavas (Coats et al., 2018) and other geomaterials. Using the average of 3 tests the highest compressive strength (44.81 MPa) was the least porous sample UNZ14 and lowest (17.69 MPa) was intermediate porosity sample UNZ1 (Fig. 5a to e, Fig. 6a, Table 4). The standard deviation and coefficient of variation of UCS were highest in anisotropic UNZ9b, and lowest in the weakest sample, UNZ1 (Table 4).

The saturated UCS tests showed that 4 of the 5 sample groups had lower saturated compressive strength than the average of the dry tests, and 3 of the 5 were lower than any of the dry tests of their respective sample group (Fig. 5a to e, Table 4), indicating a slight decrease in UCS in saturated conditions (Fig. 6b). Sample UNZ14 remained the strongest sample in compression in saturated conditions, but the most porous sample UNZ13 was the weakest of the saturated samples, unlike at dry conditions (Fig. 5a-e).

Table 4: Sample mechanical properties under dry and saturated conditions.

| Test | Environment | Sample | Identifier | Connected porosity | Strength | | | | Young's modulus | | | |
|---|---|---|---|---|---|---|---|---|---|---|---|---|
| | | | | | Measured | Average | Standard Deviation | Coefficient of variation | Young's Modulus | Average | Standard Deviation | Coefficient of variation |
| | | | | % | MPa | MPa | MPa | % | GPa | GPa | GPa | % |
| Compressive | Dry | UNZ1 | 1 | 20.44 | 17.51 | 17.69 | 0.49 | 2.77 | 4.49 | 5.46 | 1.05 | 19.20 |
| | | | 2 | 16.97 | 17.31 | | | | 6.57 | | | |
| | | | 4 | 18.23 | 18.24 | | | | 5.31 | | | |
| | | UNZ9a | 5 | 16.48 | 31.45 | 31.12 | 1.98 | 6.37 | 12.41 | 10.99 | 1.93 | 17.52 |
| | | | 8 | 12.53 | 32.91 | | | | 11.77 | | | |
| | | | 11 | 17.16 | 28.99 | | | | 8.8 | | | |
| | | UNZ9b | 1 | 16.49 | 19.21 | 22.99 | 11.86 | 51.57 | 4.58 | 6.26 | 2.95 | 47.11 |
| | | | 9 | 13.92 | 36.27 | | | | 9.66 | | | |
| | | | 10 | 20.55 | 13.48 | | | | 4.53 | | | |
| | | UNZ13 | 2 | 25.05 | 29.20 | 21.38 | 7.05 | 32.99 | 10.3 | 8.92 | 1.67 | 18.67 |
| | | | 9 | 29.59 | 19.44 | | | | 9.39 | | | |





| | | | | | | | | | | | | |
|---|---|---|---|---|---|---|---|---|---|---|---|---|
| | | | 13 | 32.32 | 15.50 | | | | 7.07 | | | |
| | | UNZ14 | 2 | 14.37 | 42.74 | | | | 14.23 | | | |
| | | | 4 | 13.18 | 47.80 | 44.81 | 2.65 | 5.92 | 15.82 | 17.21 | 3.87 | 22.50 |
| | | | 11 | 12.74 | 43.88 | | | | 21.59 | | | |
| | | UNZ1 | 12 | 18.54 | 18.31 | | | | 5.26 | | | |
| | | UNZ9a | 3 | 14.5 | 25.23 | | | | 7.99 | | | |
| | Saturated | UNZ9b | 11 | 17.01 | 22.40 | | | | 4.31 | | | |
| | | UNZ13 | 3 | 29.97 | 12.87 | | | | 4.12 | | | |
| | | UNZ14 | 3 | 13.75 | 36.53 | | | | 10.07 | | | |
| | | | 1 | 21.31 | 1.70 | | | | | | | |
| | | UNZ1 | 7 | 17.53 | 1.92 | 1.93 | 0.24 | 12.43 | | | | |
| | | | 12 | 17.38 | 2.18 | | | | | | | |
| | | | 4 | 17.16 | 1.52 | | | | | | | |
| | | UNZ9a | 6 | 8.35 | 2.57 | 1.80 | 0.68 | 37.78 | | | | |
| | | | 9 | 17.71 | 1.30 | | | | | | | |
| | | | 1 | 19.71 | 2.94 | | | | | | | |
| | Dry | UNZ9b | 4 | 18.47 | 3.69 | 3.39 | 0.40 | 11.75 | | | | |
| | | | 5 | 9.77 | 3.55 | | | | | | | |
| | | | 8 | 34.84 | 1.82 | | | | | | | |
| Tensile | | UNZ13 | 9 | 31.96 | 2.10 | 2.01 | 0.16 | 8.19 | | | | |
| | | | 14 | 30.18 | 2.11 | | | | | | | |
| | | | 2 | 13.01 | 3.70 | | | | | | | |
| | | UNZ14 | 5 | 14.52 | 2.77 | 3.01 | 0.60 | 20.01 | | | | |
| | | | 8 | 14.66 | 2.57 | | | | | | | |
| | | UNZ1 | 2 | 18.97 | 1.58 | | | | | | | |
| | | UNZ9a | 8 | 12.58 | 2.44 | | | | | | | |
| | Saturated | UNZ9b | 6 | 19.61 | 2.94 | | | | | | | |
| | | UNZ13 | 3 | 33.7 | 2.52 | | | | | | | |
| | | UNZ14 | 4 | 13.8 | 2.64 | | | | | | | |

The UTS of dry samples ranged from 1.30 to 3.70 MPa and had significant variability as a function of porosity (Fig. 6a), using the average of 3 tests the highest tensile strength (3.39 MPa) was for UNZ9b, the cataclastic sample cored perpendicular to the cataclastic fabric (note that as the sample is diametrically compressed, compression of the Brazilian disc is parallel to the plane of the fabric and the tensile rupture is also parallel; see Fig. S1) and the lowest (1.80 MPa) was for UNZ9a, the cataclastic sample cored parallel to the fabric (note that as the sample is diametrically compressed, compression of the Brazilian disc is also parallel to the fabric and the tensile fracture development is thus perpendicular; see Fig. S1), despite their similar porosity






(Fig. 5f-j, Fig. 6a, Table 4;). The standard deviation and coefficient of variation of UTS were highest in the weakest sample, anisotropic UNZ9a, and lowest in isotropic UNZ13 (Table 4).

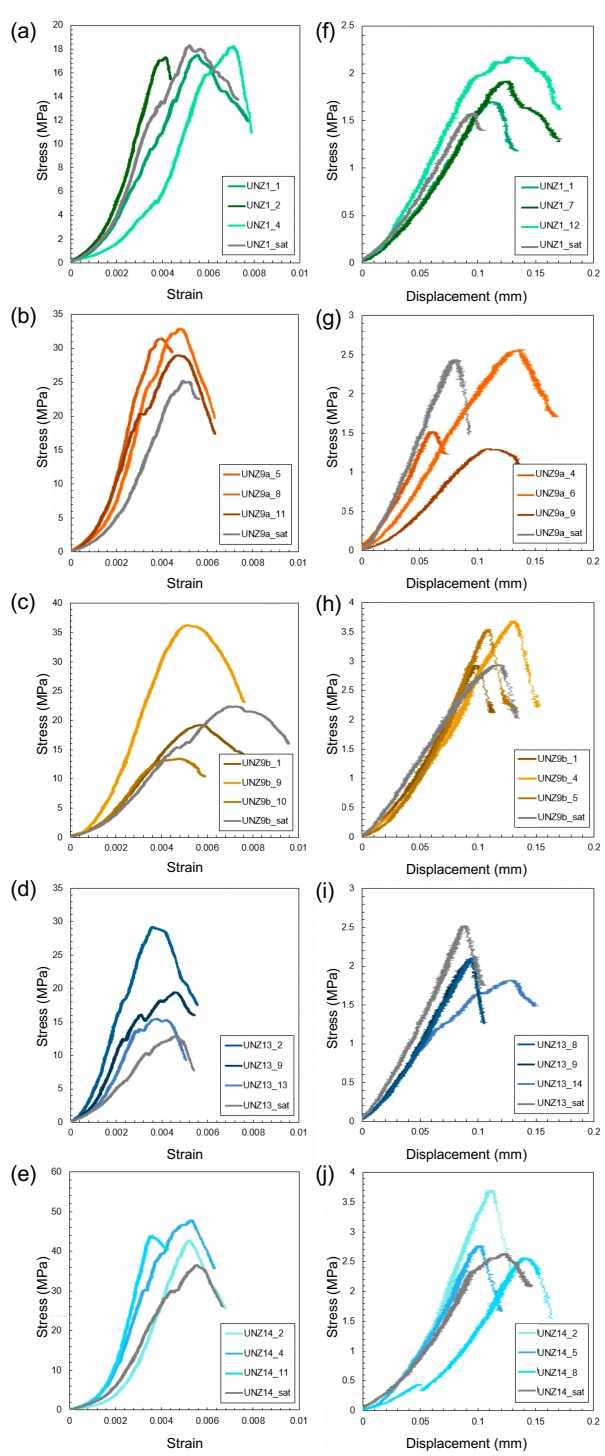





**Fig. 5: Stress-strain curves for uniaxial compressive strength (UCS) tests under dry and saturated conditions on samples (a) UNZ1, (b) UNZ9a, (c) UNZ9b, (d) UNZ13, and (e) UNZ14 and stress-displacement curves for indirect tensile strength testing (UTS) using the Brazilian disc method on samples (f) UNZ1, (g) UNZ9a, (h) UNZ9b, (i) UNZ13, and (j) UNZ14. Note the differing scales. Curves are characterised by initial portions upwards concave portions of pore closure, a linear elastic portion and transition to strain hardening damage accumulation prior to yielding and failure (stress drop).**


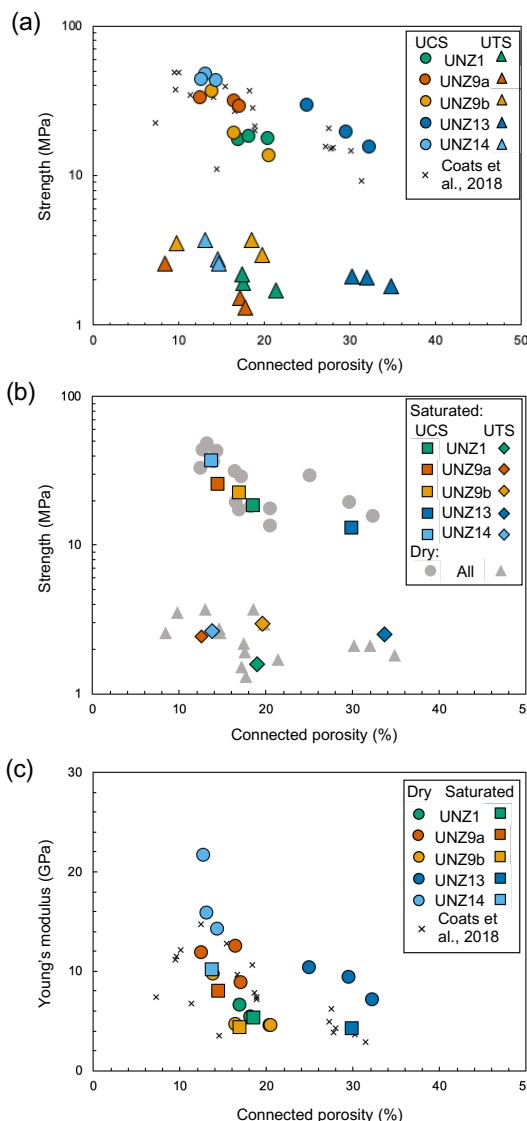

**Fig. 6: Mechanical data for: (a) dry compressive (UCS) and tensile (UTS) strength tests, plotted with comparable tests on Unzen dacite from Coats et al. (2018), as a function of connected porosity; (b) saturated tests of compressive (UCS) and tensile (UTS) strength as a function of connected porosity, with dry tests from**




**(a) plotted in greyscale for comparison; (d) Young's modulus calculated from UCS data for dry and saturated samples, plotted with comparable measurements from Coats et al. (2018), as a function of connected porosity.**

The saturated UTS tests showed that 3 of the 5 sample groups had lower saturated tensile strength than the average of the dry tests, but 1 of the 5 was higher than any of the dry tests of their respective sample group (Fig. 5f-j, Table 4), indicating no systematic change in UTS under saturated conditions (Fig. 6b). Sample UNZ9b remained the strongest sample in tension in saturated conditions, but the intermediate porosity sample UNZ1 was the weakest of the saturated samples, unlike at dry conditions (Fig. 5f -j).

### 3.4.2 Young's Modulus in dry and saturated conditions

The Young's modulus of dry samples ranged from 4.49 to 21.59 GPa, and similarly to UCS showed a broadly negative correlation with porosity (Fig. 6c) similar to previous tests on Mount Unzen lavas (Coats et al., 2018). Using the average of 3 tests the highest Young's modulus (17.21 GPa) was for the least porous, highest UCS sample UNZ14 and lowest (5.46 GPa) was for the intermediate porosity, weakest sample UNZ1 (Table 3). The standard deviation of Young's modulus was highest in the strongest sample UNZ14, and lowest in weakest sample UNZ1, yet, the coefficient of variation of Young's modulus was highest in intermediate strength, anisotropic UNZ9b, and very similar in the other samples (Table 3). The Young's modulus was systematically reduced in all saturated compression tests (Fig. 6c, Table 4).

### 3.5 Interrelation of mechanical properties

Compressive and tensile strength and Young's modulus of geomaterials depend largely on porosity, as such we examine the interrelation of these parameters to provide first-order constraints of one parameter from another.

### 3.5.1 Porosity, compressive and tensile strength

Considering each sample group, we find that UCS is between 6.8 and 17.3 times higher than UTS, with the anisotropic samples cored parallel to fabric (UNZ9a) having the highest values, and those cored perpendicular having the lowest values (UNZ9b; see Fig. S1). To compare the trends across the sample suite we first defined compressive strength ($\sigma_{UCS}$) and tensile strength ($\sigma_{UTS}$), in MPa, as a function of porosity ($\phi$), in % (for the connected porosity range 9 - 38 %). We employed least squares regressions to define empirical power law relationships (for graphical representations and appraisal of variance, see Fig. S4) of:

$$\sigma_{UCS} = 459.35\phi^{-1.016} \tag{8}$$

and

$$\sigma_{UTS} = 4.9009\phi^{-0.264} \tag{9}$$





which demonstrate that UCS reduces more significantly as a function of increasing porosity and enables estimation of UCS
and UTS for a given porosity (or porosity estimation for a given strength). We then combined these equations to define the
relationship between UCS and UTS:

$$\sigma_{UTS} = \sigma_{UCS}{}^{0.26} \tag{10}$$

showing the non-linearity of their interrelation, which is further defined by the UCS:UTS ratio as a function of porosity:

$$\frac{\sigma_{UCS}}{\sigma_{UTS}} = 93.728\phi^{-0.752} \tag{11}$$

### 3.5.2 Porosity, compressive strength and Young's Modulus

We employed the same approach to define Young's modulus ($E$) in MPa (N.B. typically given in GPa) as a function of porosity:

$$E = 82468\phi^{-0.811} \tag{12}$$


and Young's modulus as a function of compressive strength as:

$$E = 618.42\sigma_{UCS}{}^{0.7982} \tag{13}$$

showing a strong positive correlation, which can be further described by the UCS: $E$ ratio evolution as a function of porosity:

$$\frac{\sigma_{UCS}}{E} = 179.53\phi^{0.205} \tag{14}$$

In combination these relationships enable the constraint of any of the porosity, compressive strength, tensile strength and
Young's modulus from any single parameter (for graphical representations and appraisal of variance, see Fig. S4), and
moreover they provide a reasonable estimate of the range of these parameters for the variety of erupted materials, here spanning
the porosity range 9-38 % (1st to 99th percentile of the eruptive products). The modelled ranges here are UCS: 11.40-49.28
MPa; UTS: 1.88-2.74 MPa; Young's modulus: 4.32-13.88 GPa. Compared to the measured range of UCS: 13.48-47.80 MPa;
UTS:1.30 to 3.70 MPa; Young's modulus: 4.49 to 21.59 GPa.

### 3.6 Acoustic emission rate

By assuming an exponential acceleration in AE release rate we defined the maximum *a posteriori* (MAP) model parameters;
$k$, which relates to the absolute amplitude of the acceleration, and $\lambda$, the exponential rate parameter (Eq. (7), Fig. S5; after
Bell, 2018). We found that the exponential model more closely replicated the acceleration in AE rate for the tensile strength
tests, whilst compressive tests tended to have relatively high early rates of AEs inconsistent with this model (See Fig. S5).





**Fig. 7: Acoustic emission analysis data for all dry tests in compression and tension (for the dataset see Fig. S5). The exponential parameter *k* (which relates to the absolute amplitude of acceleration) is shown as function of connected porosity for (a) UCS with no systematic relationship and (b) UTS with a weak positive correlation.**





**The exponential rate parameter $\lambda$ is plotted as a function of connected porosity for (c) UCS and (d) UTS, both showing a weak positive correlation. A negative correlation is seen between $b$-value and connected porosity for (e) UCS and (f) UTS. The AE data were split into thirds to examine $b$-value evolution across first ($b_1$), second ($b_2$) and third ($b_3$) segments of deformation as a function of connected porosity, showing for (g) UCS tests and (h) UTS tests. This evolution is quantified by $\Delta b$ ($b_1$- $b_3$) which disparately shows (i) a positive correlation that spans increasing to decreasing $\Delta b$ during tests as a function of connected porosity for UCS and (j) $\Delta b$ largely decreasing, and negative correlation of $\Delta b$ with connected porosity for UTS.**

Differences between compressive and tensile tests and variability between sample groups can most effectively be described by examining the model parameters $k$ and $\lambda$. $k$ is shown as function of connected porosity for UCS and UTS tests respectively in Fig. 7a and b; showing that $k$ is typically slightly higher and spans a broader range in compression than in tension (Table S3). In compression $k$ is highest for the most porous sample, UNZ13 and lowest for intermediate porosity sample UNZ1, whilst the lowest porosity sample UNZ14 and the anisotropic samples UNZ9a and UNZ9b have intermediate values, suggesting no systematic relationship between porosity and the absolute amplitude of the acceleration of AEs. In tension, a positive correlation exists between connected porosity and $k$, with the most porous samples having highest absolute amplitude of the acceleration of AE. The scatter of $k$ within the sample groups is relatively high, with coefficients of variation of > 100 % for sample UNZ1 in compression and tension, and as low as 4.66 % for sample UNZ14 in compression (Table S3). $\lambda$ is plotted as a function of connected porosity for UCS and UTS tests respectively in Fig. 7c and d; showing distinctly higher values in tension than in compression, that shows the exponential rate parameter negatively correlates with the absolute amplitude of acceleration (see Fig. S6). In both compression and tension there is a minor positive correlation of $\lambda$ with connected porosity, and scatter is lower than for $k$, with coefficients of variation of < 30 % for all sample groups (Table S3).

To further understand the progression of damage during deformation we also examined the AEs from each test (Fig. S5), using the maximum-likelihood method of Roberts et al. (2015), we calculated the $b$-value for each experiment (above the cut off amplitude of -3.3). $b$-value is the negative gradient of the slope of amplitude-frequency distribution, therefore a lower $b$-value is an indication of a greater proportion of higher amplitude events. We found that the $b$-value has a negative correlation with connected porosity for both UCS and UTS tests (Fig. 7e-f), and that it was higher in tension than compression. Repeatability within sample groups was typically good in both compression and tension, with coefficients of variation < 14 % for all groups (Table S3). $b$-value has a poor positive correlation with $\lambda$ and minor negative correlation with $k$ (Fig. S6).

In addition, the $b$-value was determined for each third of every test to examine evolution during deformation (Fig. 7g-h). As with the $b$-values for the whole tests, the $b$-values for each third had a negative correlation with porosity, yet for the tests in compression the sensitivity of $b$-value to connected porosity seemed to increase during the tests (slope of $b_3$ is steepest, Fig. 7g) whilst in tension the sensitivity of $b$-value to connected porosity seemed to decrease (slope of $b_3$ is shallowest, Fig. 7h). To





examine this further we defined $\Delta b$, the difference between the first and final thirds ($b_1$ - $b_3$). This analysis showed that in compression $\Delta b$ correlates positively with connected porosity and transitions from negative (increasing $b$-value during deformation) to positive (decreasing $b$-value during deformation) as a function of porosity. In other words, that $b$-value

increased during tests on low porosity samples (negative $\Delta b$) but decreased for more porous samples (positive $\Delta b$; Fig. 7i). However, in tension $b$-value almost always decreased (positive $\Delta b$), and this $\Delta b$ negatively correlated to connected porosity, such that $b$-value reduction during deformation was more significant (high $\Delta b$) at lower porosity (Fig. 7j).

### 3.7 Coda wave interferometry

We examined the deformation induced during dry and saturated compressive and tensile tests using active pulsing across paired PZTs on opposing edges of the samples (see Fig. S1). Received bursts were stacked and coda wave interferometry (CWI) was applied to the stacks following the method of Lamb et al. (2017) to calculate the variance of the travel time perturbation, and thus to calculate relative change in seismic wave velocity during the experiments (velocities are typically higher for denser materials). Velocity evolution for all experiments as a function of test duration, normalised to 100 % at the time of sample

failure are plotted in Fig. 8.

Under compression (Fig. 8a-e) the dry samples show velocity change that fluctuates about 0 for at least the first 50 % of time to failure in the tests, after which velocity reduction is more pronounced for some tests than for others; the least porous sample, UNZ14 appears to have the strongest evolution. The saturated samples in compression however fluctuate about 0 for the entire

duration, showing no velocity change induced by damage evolution during testing. During tensile tests (Fig. 8f-j) a similar behaviour is observed for the dry samples, except velocity reduction appears to onset later, around 60-70 % of time to failure, and the most porous sample, UNZ13 appears to be most influenced. In tension the saturated samples again fluctuate about 0 for the entire duration of the tests and show no velocity change induced by damage evolution.

To enable systematic comparison between tests we devised an approach whereby a linear fit with forced intercept (at 0-0) was applied to the velocity change data using a least squares approach, and the intercept of the line with the end of the test (time to failure = 100 %) was defined as the magnitude of the velocity change. [We acknowledge that the outcome may lead to underestimation of velocity change (for some tests even resulting in a false positive velocity change as the porosity reduction typically occurs only in the latter stages of the tests and thus may be outweighed by fluctuations). We also acknowledge that

this approach may not capture the subtleties (for example timing) of material damage accumulation, but felt it a more robust approach than selecting the maximum velocity change which may represent a data spike.] Due to the high scatter of the data generated by CWI we posit that such an approach is required for comparisons to be made (values are provided in Table S4, yet we suggest that their utilisation is for the purpose of exploring trends rather than quantitative assessment). We plot the velocity change defined as such against porosity for compressive and tensile tests in Fig. 9a and b. For the dry compression

tests we see no systematic variation in velocity change as a function of porosity, and for the saturated tests, as observed in the





velocity change traces through time (Fig. 8) we see almost no variability (Fig. 9a). For the dry tensile tests we see a minor negative correlation between connected porosity and velocity change, or in other words, a greater velocity reduction in more porous samples, and again for saturated samples we see almost no variation (Fig. 9b).

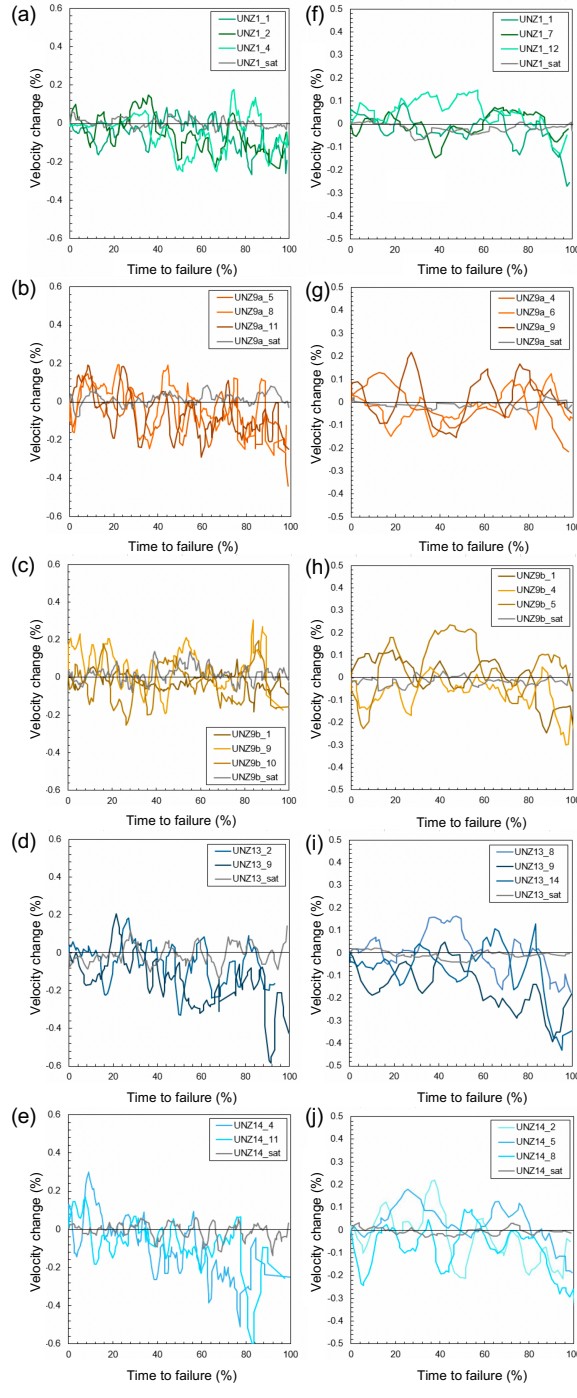





**Fig. 8: Coda wave interferometry data presented as velocity change as a function of time to failure (normalised to test length) for dry and saturated (sat) conditions: UCS tests on samples (a) UNZ1, (b) UNZ9a, (c) UNZ9b, (d) UNZ13, and (d) UNZ14; and UTS tests on samples (f) UNZ1, (g) UNZ9a, (h) UNZ9b, (i) UNZ13, and (j) UNZ14. For UCS tests (a-e) velocity reduces after ~ 50 % time to failure for dry tests, but continues to fluctuate about 0 (no velocity change) throughout saturated tests. For UTS tests (f-j) velocity reduces after 60-70 % time**
**to failure for dry tests and also continues to fluctuate about 0 for the saturated tests.**

We additionally compared velocity change to *b*-value, finding a relatively good correlation whereby lower *b*-value accompanied larger velocity reductions in compression (Fig. 9c) and to a lesser but still observable extent in tension (Fig. 9d). To further explore the relationship between velocity change and acoustic emissions we compared velocity change to $\Delta b$ (the
difference between the *b*-value of the first and final thirds of the tests. For compression tests larger reductions in *b*-value (higher $\Delta b$) corresponded to larger reductions in velocity (Fig. 9e). For tension tests the relationship was less clear, and perhaps showed a poor counter-correlation, whereby larger reductions in *b*-value (higher $\Delta b$) corresponded to less significant velocity changes (Fig. 9f).

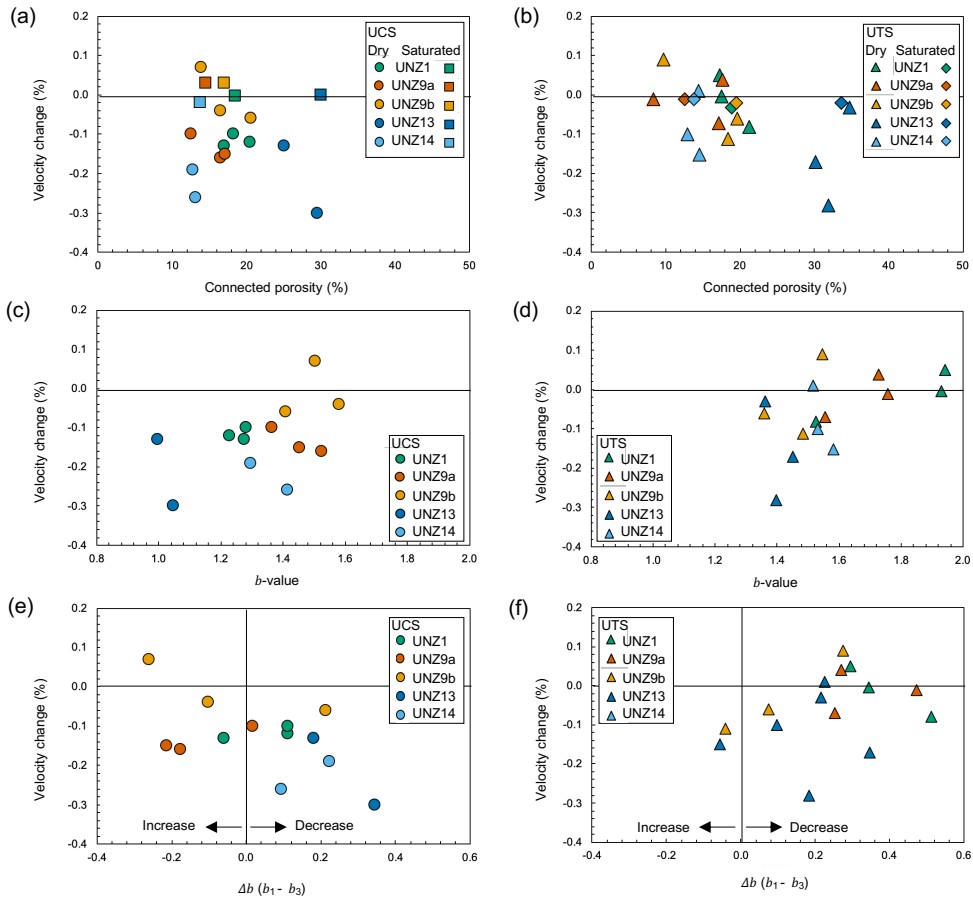



**Fig. 9: Magnitude of velocity change during mechanical testing compared to connected porosity for dry and saturated (a) UCS and (b) UTS tests, showing no correlation for dry UCS tests, a minor negative correlation for dry UTS tests and no change for saturated tests. Magnitude of velocity change compared to _b_-value calculated from acoustic emission monitoring, showing a weak positive correlation for dry (c) UCS and (d) UTS tests. Magnitude of velocity change compared to _Δb_-value ($b_1$- $b_3$) during dry (e) UCS and (f) UTS tests,**
**showing disparate correlation in compression and tension**.

## 4 Interpretation and discussion

### 4.1. Relationships between physical and mechanical attributes

The density and the connected, isolated and total porosities for the Mount Unzen lavas studied here match previously
constrained values for the 1990-95 dome eruption (Coats et al., 2018; Cordonnier et al., 2008; Hornby et al., 2015; Kueppers et al., 2005; Wallace et al., 2019). Averages of the 5 sample groups showed a density range from 1.54-2.40 g.cm$^{-3}$, total porosity range from 16.05 to 36.46 %, isolated porosity range from 0.39 to 5.37 % and connected porosity range from 13.69 to 33.13 %, ranking the samples as follows from least to most porous: UNZ14, UNZ9a, UNZ9b, UNZ1, UNZ13 (Fig. 4a, Table 1). The anisotropic samples (UNZ9a and b) have both a higher degree of connectivity and higher degree of variability of porosity than
the isotropic samples, indicative of a tortuous stress-strain history during their genesis.

The range of all 1065 permeability measurements on cores and discs spanned 1.54 x10$^{-14}$ to 2.67 x10$^{-10}$ m$^2$ with coefficients of variation of over 259 % within a single sample, suggesting significant rock heterogeneity on the scale of the samples (Fig. 4b, Table 1). Permeability is largely dictated by porosity, but for a given porosity (considering the average for each core or
disc), the permeability can span > 2 orders of magnitude, indicative of the variability in the porous network geometry; for example, at the scale tapped here, permeability is higher for the macroscopically isotropic samples UNZ1 than the macroscopically anisotropic samples UNZ9 of similar porosity. This suggests the area sampled by the measurements (conducted via pulse decay through an 8 mm circular aperture) is more sensitive to pore geometry than to larger-scale heterogeneities, supported by the observation that these permeability measurements do not distinguish between anisotropic
samples cored parallel or perpendicular to fabric (Table 1; though slight clustering of samples in porosity-permeability space can be seen in Fig. 4b, due primarily to slight differences in porosity). We conducted a further 379 unconfined permeability measurements on cut, planar block surfaces of macroscopically anisotropic UNZ9 and dense, relatively isotropic UNZ14, which revealed similar average permeabilities to the measurements made on cores and discs, but higher standard deviations and coefficients of variation (for each block), highlighting a degree of variability in the blocks not captured at the core and
disc scale (Table 2).

Confined permeability measurements revealed permeability reduction as a function of effective pressure, highlighting the greatest sensitivity in the lowest permeability samples (Fig. 4c), potentially due to the relative ease of closing high aspect ratio





fractures compared to more equant pores (e.g. Lamur et al., 2017; Zimmerman, 1991; Kennedy et al., 2020; Gueguen and
Dienes, 1989; Griffiths et al., 2017). Similar porosity-permeability relationships to those in unconfined measurements were
noted for the five sample groups, including the higher permeability of isotropic samples compared to anisotropic samples with
similar porosities. In addition, it was noted that the cores cut parallel to the cataclastic fabric (UNZ9a) were more than an order
of magnitude more permeable than those cut perpendicular (UNZ9b), which was not observed in the unconfined permeability
measurements performed on the cores, discs and planar block surfaces. This suggests that the confined measurements, with
the larger sampling volume, tap a different scale of heterogeneity, capturing the impact of the cataclastic banding visible in the
specimens (Fig. 2). In addition, preferential closure of the microfractures observed in the dense layers in thin section (Fig. 3)
upon confinement heightens the permeability anisotropy in already anisotropic samples; in particular, the denser layers in
perpendicular-cut UNZ9b serve to block fluid flow (fluid may not circumnavigate the dense layers), whereas the same dense
layers running parallel to fluid flow have negligible influence on fluid transmission, which is primarily hosted in the porous
layers. This suggests that the impact of anisotropy on fluid flow in volcanic systems may be more significant at depth than in
shallow (~ unconfined) settings. It also highlights the need to use measurements made at comparable effective pressures to
compare samples' permeabilities. Furthermore, the two types of permeability measurements (unconfined with small sampling
volume, versus confined with larger sampling volume) highlight the importance of the scale of examination when defining the
physical attributes of materials, the small sampling volume of the unconfined measurements is able to distinguish between
pore geometry (e.g. crack versus vesicle dominated in the anisotropic and isotropic samples, respectively) but unable to
discriminate the orientation of anisotropic fabrics seen to dominate the UNZ9 samples, thus in the description of permeability
the scale should always be defined.

The uniaxial compressive strength (UCS) of the samples was primarily controlled by porosity (Fig. 5, Fig. 6, Table 4) as has
been noted for Unzen lavas (Coats et al., 2018), similar volcanic rocks (Harnett et al., 2019a; Heap et al., 2014a; Schaefer et
al., 2015) and a broad range of geomaterials (Paterson and Wong, 2005). Under dry UCS conditions the lowest porosity UNZ14
was the strongest sample group (44.81 MPa) and intermediate porosity sample UNZ1 was weakest (17.69 MPa). Weak UNZ1
is notably more fracture-dominated than the most porous, vesicle dominated UNZ13, which is stronger (21.38 MPa)suggesting
that as well as absolute porosity, the geometry of pore space is influential on rock strength (e.g. Bubeck et al., 2017; Griffiths
et al., 2017). As the pore geometries of volcanic rocks are highly variable due to their complex formation histories, it is thus
important to understand the microstructural characteristics in order to estimate strength. Furthermore, the results show that
UCS is dependent on sample-scale anisotropy, with rocks compressed parallel to the cataclastic fabric (of denser and more
porous bands) being stronger than those perpendicular (UNZ9a compared to UNZ9b). The influence of anisotropic fabrics on
UCS of volcanic rocks has been noted previously (e.g. Bubeck et al., 2017), with maximum strength typically considered to
be when anisotropy aligns at ~30º from application of the principal stress, though the specific properties of fabrics means this
may not always be the case. The standard deviation and coefficient of variation of UCS were also highest in the anisotropic
sample (UNZ9b), suggesting anisotropy further fuels variability in strength of lava domes and volcanic edifices.





Similarly the Brazilian disc method showed that UTS is primarily controlled by porosity (Fig 5, Fig. 6, Table 4), as has been

noted for other volcanic rocks; cold (e.g. Harnett et al., 2019a), hot (Hornby et al., 2019) and fragmented by pore overpressure (Spieler et al., 2004). The average UTS of each group of dry samples revealed that the strongest was UNZ9b (3.39 MPa), the cataclastic sample cored perpendicular to the cataclastic fabric (and fractured in tension parallel to fabric) and weakest was UNZ9a (1.80 MPa), the cataclastic sample cored parallel to the fabric (thus fractured in tension perpendicular) despite having equivalent porosities. Thus tensile strength is potentially more sensitive to anisotropy than compressive strength, being

controlled by the weakest element that traverses the material perpendicular to the applied tensile stress (e.g. Lydzba et al., 2003). The weak anisotropic samples (UNZ9a) also had the highest standard deviation and coefficient of variation of UTS, whilst lower values were seen in the stronger and isotropic samples. Similarly to the results of UCS testing, this suggests that anisotropic samples further add to strength variability of volcanic materials that can promote structural instability.

Saturation served to decrease the strength of 4 out of 5 sample groups in UCS and for 3 out of 5 in UTS, though in UTS one group was also stronger saturated, as such the effect of saturation is interpreted to slightly differ in compression and tension (Fig. 5, Fig. 6, Table 4). In compression, pore fluid serves to impact material poro-mechanically and chemically including stress corrosion by capillary action at the fracture tip, reduction of the friction angle (due to the lubricating influence of water) and, depending on the ability for the fluid to escape, the saturated pores may develop pressure (called the crack splitting tensile

stress) as the sample is compressed; as such the presence of water enhances the growth of fracture networks and typically weakens material (e.g. Grgic and Giraud, 2014). Yet, in tension the influence of water may be different, the tendency for Brazilian disc tests to fail rapidly by the generation of a single pervasive fracture (compared to the creation, propagation and coalescence of many fractures towards shear failure in compression) may limit the material surfaces available for stress corrosion, the lubricating effect of the water may have a lesser effect in this stress field (compared to in compression) and

pores are unlikely to develop heightened pressure (except potentially during initial stage of loading during inelastic "crack-closure") that can enhance the efficiency of ruptures. Indeed, fracture propagation in saturated rocks in tension has been seen to be significantly slower than in dry conditions (Wong and Jong, 2014), which could be a result of reduced efficiency of capillary action in growing pore space filled with a finite water volume (as noted in other materials; e.g. Smith, 1972). The results here are inconclusive as to whether water saturation increases or decreases tensile strength of porous volcanic rocks,

and further study would be required to draw a conclusion. It is however pertinent to note that the impact of the pore fluid is likely a result of interplay between deformation rate and permeability, and that in volcanoes strain rates can span > 10 orders of magnitude, thus a full description of the role of saturation on strength would also require characterisation of its rate dependence.

Young's modulus of the materials tested mimicked patterns observed for strength, with the highest Young's modulus for the least porous, strongest sample, UNZ14 (17.21 GPa) and lowest for the weakest sample, UNZ1 (5.46 GPa; Fig. 6c, Table 4).



Young's modulus was systematically reduced by saturation, as has been noted in previous studies (e.g. Makhnenko and Labuz, 2016). As compressive and tensile strength and Young's modulus scale primarily with porosity, we provide relationships to estimate each parameter from one another (Eq. (8)-(14), Fig. S4). Whilst this approach lacks the precision of more nuanced
micro-mechanical solutions (e.g. Paterson and Wong, 2005 and references therein), it has the benefit of being broadly applicable to the range of materials against which it is calibrated, i.e. here porphyritic dacites in the porosity range from 9 to 38 %, without relying upon microstructural characterisation of dominant pore size and/ or crack length. For example, previous work (Coats et al., 2018; Heap et al., 2014c) has shown that the evolution of pore geometry in volcanic rocks in this porosity range renders end-member solutions such as the pore-emanated crack model (Sammis and Ashby, 1986) and wing-crack model
(Ashby and Sammis, 1990) ineffective without a weighted solution that incorporates both. UCS is often approximated at 10 times higher than UTS. Here, we found that UCS is between 6.8 and 17.3 times higher than UTS, with the modelled results highlighting that the ratio typically decreases with increasing porosity (as UTS is less sensitive to increasing porosity), though for the samples tested the anisotropic rocks account for the maximum (cored parallel to fabric, UNZ9a) and minimum (cored perpendicular, UNZ9b) UCS:UTS ratio. This suggests that pore geometry and connectivity has a significant control on the
UCS:UTS ratio, complementing results of Harnett et al. (2019a) who found higher UCS:UTS ratios for stronger, less permeable materials, and cautioned against using a constant ratio in numerical models of lava domes.

Young's modulus increases non-linearly with UCS, and is between 275-375 times higher than UCS, increasing with increasing porosity, suggesting Young's modulus is slightly less sensitive to increasing porosity than compressive strength. The modelled
relationships are able to capture the range of material characteristics reasonably well, giving, for the porosity range 9-38 %, a range of: UCS of 11.40-49.28 MPa (compared to measured range of 13.48-47.80 MPa); UTS of 1.88-2.74 MPa (compared to measured range of 1.30 to 3.70 MPa) and Young's modulus of 4.32-13.88 GPa (compared to measured range of 4.49 to 21.59 GPa). The modelled relationships fail to capture endmembers of measured results (for which the value ranges are higher), heightened by the inclusion of anisotropic samples here (e.g. anisotropic samples have the highest and lowest UTS
measurements despite having equivalent porosity). This highlights that even using relationships defined directly from laboratory measurements leads to an underestimation of the range of mechanical heterogeneity, and if modelling a lava dome or volcanic edifice then previous studies have reported that upscaling of mechanical properties may see intact rock strength values further weakened by as much as 80-97 % (Thomas et al., 2004; Walter et al., 2019). This is due to large scale heterogeneities and other phenomena that may act locally such as alteration (weakening or strengthening), temperature
heterogeneity (weakening or strengthening), unconsolidated ash/tephra layers (weakening), confining pressure (strengthening), saturation (weakening) etc. Yet the utilisation of such laboratory-constrained physical and mechanical property ranges rather than fixed and/ or estimated values is necessary to model the complexity of structures in volcanic complexes (e.g. Husain et al., 2014).





A change in stress state of a material, even without failure, can impact the stability of volcanic edifices or lava domes. For example, dilation can result in enhanced permeability (and permeability anisotropy) which may allow pressurised fluid or gas to infiltrate or escape (enhancing or reducing instability risk). Deeper in volcanic systems (under confinement) dynamic closure of fractures (e.g. Watanabe et al., 2008) or compaction of weak or unconsolidated material can reduce permeability (Kennedy et al., 2020) that can drive the development of overpressure or shift the stress field in overlying rocks that can also affect

stability. Understanding the progression of microstructures during deformation, both in the laboratory and at volcanoes, can be enhanced by geophysical monitoring.

### 4.2. Signals of heterogeneous material deformation

Passive and active acoustic emission monitoring was employed to track the evolution of materials during deformation. The

acceleration of AE rate in the tensile strength tests were well described by the exponential model, whereas compression tests had elevated AE rates during the early phase of deformation. The range of the absolute amplitude of the acceleration ($k$) and exponential rate parameter ($\lambda$) are distinct for compressive and tensile tests; $k$ is higher in compression whilst $\lambda$ is higher in tension (Fig. 7). These results suggest that failure forecasting may be more effective in tensile regimes, though forecasting windows may be shorter (due to the relatively later onset of cracking events) and the translation of this observation to

deformation at crustal scale may not be straightforward. We also find that $\lambda$ increases with increasing porosity in both compression and tension, suggesting that more porous materials may facilitate more rapid coalescence of fractures, whilst $k$ varies less systematically (Fig. 7).

Examination of $b$-values elucidates further distinctions in the physical evolution between samples and between deformation in

compression and tension. Specifically, $b$-value is up to 0.5 higher for UTS than UCS for a given material (Fig. 7e-f). A lower $b$-value indicates a higher proportion of higher amplitude events. Scholz (1968) found that for a wide range of rock types, $b$-value was higher during pervasive, ductile (compactant) deformation than for localised brittle (dilatant) deformation, and comparably, Lavallée et al. (2008) showed that $b$-value decreased with increasing applied stress or strain rate, which enhanced localisation (resulting in more, larger events) in vesicular, crystalline lavas. Previous work on regional seismicity has shown

that $b$-value is higher in extensional than in compressive regimes (Schorlemmer et al., 2005), and moreover, that $b$-value reflects both the spatial distributions of fracturing events and the focal mechanism, such that $b$-value is elevated at low stress intensities. The higher $b$-value for our tensile tests (compared to compression) mirrors these prior observations and can be attributed to contrasting differential stresses in the two regimes (e.g. Scholz, 1968).

Tests on variably porous sintered aphyric suspensions showed $b$-value increasing as a function of heterogeneity (~ pores, in their study; Vasseur et al., 2015). To our knowledge no such study has previously been performed on a series of variably porous natural volcanic rocks, or in tension. We demonstrate that $b$-value has a negative correlation with connected porosity for both UCS and UTS tests (i.e., failure of porous rocks results in lower $b$-value, than for dense rocks; Fig. 7). At first, this



observation appears at odds with previous studies (e.g. Vasseur et al., 2015; Scholz, 1968) but examination of microstructures
reveals the cause. Previous work has shown that pore size, geometry and distribution all impact stress intensity (e.g. Meredith
and Atkinson, 1983), with larger, more closely clustered pores leading to higher stress intensities, which in turn has a strong
negative correlation with $b$-value (e.g. Ribeiro, 2012). In UCS the highest $b$-values are in cataclastic banded samples UNZ9b
and UNZ9a; despite the overall low porosity of these samples, large areas are covered by granular bands that facilitate
numerous fracture nucleation sites and low stress intensities. This is followed by samples UNZ14, then UNZ1, both of which
have low to intermediate porosity including elongate narrow fractures that enable a substantial number of small AE events by
shear displacement during deformation. Finally, the lowest $b$-value is in the most porous sample UNZ13, but this sample has
a notable absence of microfractures, and is instead dominated by large and tightly clustered rounded to sub-rounded pores
which serve to increase stress intensity and hence reduce $b$-value. In tension (UTS) a similar progression is seen to
compression, except that the samples UNZ1 and UNZ14 which have long, fine microfractures shift to yet higher $b$-values,
suggesting that in tension such pre-existing fracture networks dominate the deformation response. The cataclastic samples
UNZ9a and UNZ9b maintain high $b$-values in tension, which show larger differences between those cored parallel and
perpendicular to banding than in compression, as would be anticipated from their contrasting strengths in tension (Fig. 6).
Finally, the UNZ13 sample also has the lowest $b$-value in tension. Thus across the porosity range tested here (13.69 to 33.13
%), pore size, geometry and distribution seem to have a more dominant control on $b$-value than absolute porosity, verifying
previous observations that stress intensity has the primary control on $b$-value (Ribeiro, 2012 and references therein).

A number of studies have indicated that $b$-value decreases during deformation (in compression) on the approach to failure as
damage localises (Vasseur et al., 2015; Lockner, 1993; Meredith et al., 1990; Main et al., 1992). We tracked the evolution of
$b$-value by splitting each test into thirds ($b_1$ to $b_3$) of equal time interval. In compression $\Delta b$ ($b_1$- $b_3$) correlates positively with
connected porosity, and transitions from negative $\Delta b$ (increasing $b$-value during deformation) at low porosity to positive $\Delta b$
(decreasing $b$-value during deformation) with increasing porosity. The unusual observation that $b$-value increases during
deformation is largely observed for the anisotropic samples, which could be due to increasing levels of compaction in the
cataclastic bands as stress accrues. In tension, $b$-value almost always decreased (positive $\Delta b$) throughout deformation as in
previous studies (e.g. Lockner, 1993; Main et al., 1992), and the magnitude of $\Delta b$ negatively correlated to connected porosity
(lower porosity samples had the biggest reduction in $b$-value, i.e. highest $\Delta b$). Whereas the acceleration of AE events can
present difficulties in precursory detection of failure in materials in nature due to the necessity to establish a reference point
(or baseline) for each unique material in each part of a system under stress, monitoring dynamic changes in $b$-value (for
example, using set time windows) may be one of the most robust indicators of precursory activity.

In addition to monitoring passive AEs we also used active pulsing and applied coda wave interferometry (CWI). In a scenario
where AEs are also being produced by deformation during the experiments this method offers an alternative, potentially more
robust approach, to the direct measurement of pulse arrival times to measure velocity change . The coda of a wave is the section





after the directly arriving phases, and, in laboratory scale rock samples comprises surface waves and waves that have repeatedly scattered (reflected within) the medium (Grêt et al., 2006; Singh et al., 2019). Where conventional approaches to measuring first arrivals are highly sensitive to local heterogeneities and thus may not accurately represent bulk material properties, CWI effectively samples the whole material multiple times, a process which provides a robust constraint of bulk properties and amplifies even very minor temporal changes compared to direct arrivals (Singh et al., 2019; Snieder et al., 2002; Hadziioannou et al., 2009; Griffiths et al., 2018). We identified velocity reductions during mechanical testing in both compression and tension, the magnitude of which is greater in more porous samples in UTS but appears independent of porosity in UCS (Fig. 9). Notably in compression the least porous sample exhibits the biggest velocity reduction, whilst in tension it is the most porous sample which is most significantly impacted. This distinction is likely due to the contrasting stress fields generated in the compression and Brazilian disc setups, which generate fractures parallel to the principal applied stress that cause velocity reduction. In tension velocity reduction began later during the tests (at 50-70 % time to failure) compared to in compression (at ~ 50 % time to failure), an observation that mirrors the AE release rates that are exponential for tests in tension but below exponential in compression.

The velocity reductions identified by CWI scale to both $b$-value and $\Delta b$ (Fig. 9). In both compression and tension, lower $b$-values were accompanied by larger velocity reductions, which both indicate the development of pervasive fractures. For compression tests larger reductions in $b$-value (higher $\Delta b$) corresponded to larger reductions in velocity. Tension tests showed a poor counter-correlation (higher $\Delta b$ corresponded to smaller velocity reductions), which may be due to a number of factors controlling velocity during Brazilian disc tests, such as the late occurrence of coalescing or pervasive fractures or distribution of dilation in some areas countered by compaction in others.

CWI has been used to monitor and detect subtle changes in the degree of water saturation of rocks (Grêt et al., 2006), yet, here we show that in a material that is deforming, saturation may in fact obscure damage accumulation; in our experiments the degree of saturation remained constantly high throughout deformation, and no velocity change was detected even during the visible creation of fractures. Such results are important for potentially saturated volcanic systems, where damage accumulation (e.g. Snieder et al., 2006; Griffiths et al., 2018) or source migration (Lamb et al., 2015) that could otherwise be monitored by CWI might be obscured by constant saturation. Alternatively, enhancement of the permeable porous network by fracturing that would allow fluid to drain or infiltrate new areas may also overprint structural or source evolution due to the sensitivity of CWI to saturation level (Grêt et al., 2006). Thus, it is vital that all such variables (i.e., evolving material properties, source migration, source mechanism, degree of saturation) be considered in the interpretation of CWI results, and laboratory experiments can elucidate their relative impact across a suite of controlled conditions.

**5 Conclusions**



Mount Unzen is a primarily dacitic volcano located in Shimabara Peninsula, Japan. The Heisei-Shinzan lava dome that forms the current summit continues to pose a collapse hazard. During a field campaign in 2015 we selected a number of porphyritic samples with a range of porosities and fabrics (isotropic and anisotropic) from block-and-ash deposits to measure and compare the physical and mechanical properties. The samples tested span a porosity of 9.14 to 42.81 %, and permeability of $1.54 \times 10^{-14}$ to $2.67 \times 10^{-10}$ $m^2$ (from 1065 measurements on rock cores). For a given porosity the permeability varies by > 2 orders of magnitude. Macroscopically anisotropic samples typically have lower permeability than isotropic samples of similar porosity. Permeability measurements made under confinement revealed that the lowest permeability samples were most sensitive to effective pressure, interpreted to result from preferential closure of crack-like pores compared to more equant vesicles. Permeability measurements made on planar block surfaces of an isotropic and anisotropic sample ranged from $1.90 \times 10^{-15}$ to $2.58 \times 10^{-12}$ $m^2$ (from 379 measurements), and within a sample suite standard deviation and coefficient of variation were higher than the measurements made on sample cores, highlighting a degree of heterogeneity in the rock samples which is not captured at conventional sample core scale. Our permeability measurements highlight the importance of detailing the scale (sample volume) of measurements, the fluid medium used and the effective pressure conditions in the use of permeability values.

The uniaxial compressive strength (UCS) of the 5 sample groups ranges from 13.48 to 47.80 MPa, and tensile strength (UTS) using the Brazilian disc method ranges from 1.30 to 3.70 MPa. Although porosity has a primary control on strength, we found that at similar porosities, crack-dominated lavas are weaker than vesicle-dominated ones. Saturation decreases UCS, but the impact on UTS is variable. UCS is between 6.8 and 17.3 times higher than UTS, with anisotropic, cataclastically banded samples cored parallel and perpendicular to fabric presenting as each end member. The orientation of the banded samples had a more significant impact on tensile strength than compressive strength considering a principal applied stress parallel and perpendicular to fabric. This was interpreted to be due to the wholesale failure being caused by a through-going fracture that could be hosted solely within the weaker cataclastic layers, whereas the shear failure of the samples under UCS traversed strong and weak layers to fail (as such, the weakest orientation for banded samples in compression is likely to be inclined). Young's modulus of dry samples ranged from 4.49 to 21.59 GPa and was systematically reduced by saturation. The interrelation of porosity, UCS, UTS and Young's modulus were defined by a series of empirical relationships that facilitate the estimation of the range of each physical or mechanical parameter from another. Whilst this approach lacks a micromechanical basis it is a useful tool to generalise the attributes of a particular material and may prove to be particularly beneficial as input parameters for the modelling of volcanic systems.

Acoustic emissions were monitored during deformation and acceleration was assessed by fitting Bayesian Poisson point process models to define the maximum *a posteriori* (MAP) model parameters, $k$ (which relates to the absolute amplitude of the acceleration) and $\lambda$ (the exponential rate parameter). The exponential model had a good fit to tensile strength tests, but compressive tests tended to have relatively high early rates of AEs: $k$ was typically higher in compression and spanned a





980     broader range, but did not vary systematically with porosity, whereas in tension *k* increased with increasing porosity; *λ* was higher for tension tests, negatively correlated with *k*, and increased with increasing porosity in both compression and tension.

The frequency-amplitude distribution of the AEs from each test defined the *b*-value. We found that *b*-value is higher in tension than compression tests, indicating a higher proportion of higher amplitude events in compression. *b*-value reflects both the

985     spatial distribution and focal mechanism of fracturing events, with high *b*-values resulting from low stress intensities, as such, we attribute the different ranges in tension and compression to contrasting differential stresses in the two regimes (e.g. Scholz, 1968). This observation matches work on regional seismicity that showed higher *b*-values in extensional rather than compressive regimes (e.g. Schorlemmer et al., 2005).

990     We found that *b*-value has a negative correlation with connected porosity for both UCS and UTS tests. We interpret the difference in this result compared to previous work (e.g. Scholz, 1968) in which *b*-value increased as a function of heterogeneity (c.f. porosity; Vasseur et al., 2015), to result from the complex and contrasting porous networks in our samples, that control the stress intensity (e.g. Meredith and Atkinson, 1983). Moreover, the presence three phases (glass, crystals, pores) in Unzen lavas prevents the simplification of porosity to heterogeneity; phase contacts serve as nucleation sites for the initiation

995     of fractures (Lavallée et al., 2008). Large, closely clustered pores, such as in our most porous samples, cause higher stress intensities (compared to micro-cracks) which in turn results in lower *b*-values (e.g. Ribeiro, 2012). Our highest measured *b*-values in compression were in cataclastic banded samples, whose granular layers facilitated low stress intensities and ample fracture nucleation sites. In tension, samples with long, fine microfractures that promoted distributed deformation also had high *b*-values.

000

*b*-value evolution during deformation tests has previously been used as a proxy for damage state, with *b*-value typically decreasing during deformation as damage becomes increasingly localised prior to material rupture (e.g. Main et al., 1992). To examine this we defined *Δb*, and found that the majority of samples had positive *Δb* values, i.e. *b*-value decreased between the first and final thirds of each test as strain localised in the approach to failure. A few samples, specifically the cataclastically

005     banded samples in compression, showed increasing *b*-value during deformation, which likely resulted from pervasive ductile damage in the porous cataclastic bands (also indicated by the high absolute *b*-values). We also found that *Δb* increased with increasing porosity, i.e. higher porosity samples suffered greater reductions in *b*-value, during compression tests. In tension, the opposite was true, i.e. lower porosity samples suffered greater reductions in *b*-value, and yet in all tests an evolution of *b*-value was indicative of approaching failure. Unlike tracking the acceleration of AE events which relies on the establishment

010     of a baseline or reference point, tracking dynamic changes in *b*-value using set time windows may be one of the most robust indicators of changing activity.



Using coda wave interferometry of active acoustic emission pulsing we identified velocity reductions during mechanical testing in compression and tension, the magnitude of which is greater in more porous samples in UTS but independent of porosity in UCS. Typically, the largest velocity reductions in both compression and tension were associated with the lowest $b$-values (indicative of relatively large, localised fractures). In compression tests comparison of velocity change to $\Delta b$ also revealed that the largest reductions of velocity corresponded to the biggest drops in $b$-value, indicating high susceptibility to damage. In tension, there is a poor opposite correlation, which we posit relates to the complex geometry of Brazilian disc tests, whereby certain areas are compacted whilst others dilate, and the relative susceptibility of crack-dominated versus vesicle-dominated materials to the distribution of stresses. We propose that CWI is a more robust measure of bulk material properties than traditional methods utilising first arrivals as the method samples the whole material and thus is more representative than the measure of a single raypath. Yet, we caution that care should be taken in the interpretation of coda in the case of multiple simultaneous dynamic changes as we found that water saturation in our tests obscured the observation of velocity reduction by accrual of damage. In naturally wet volcanic environments damage accumulation, fluid circulation and migrating or evolving seismic sources may all overprint one another, and thus laboratory experiments may be valuable in elucidating the competing control of these variables on monitored seismic signals.

The extensive physical and mechanical results presented here demonstrate the complexity of volcanic materials, even when derived from a single eruption. The work highlights that heterogeneity and anisotropy on a sample scale not only enhance variability in physical and mechanical properties at volcanic systems, but also have a defining role in the channelling of fluid flow and localisation of strain that dictate a volcano's hazards and the geophysical indicators we use to interpret unrest at persistently active volcanic complexes such as Mount Unzen.

**Code, data and sample availability**

Supplementary data are available in the Supplementary Figures S1 to S6 and Supplementary Tables S1 to S5. The script for the Young's modulus calculation is freely available on GitHub (Coats, 2018). Further data, scripts and information can be obtained upon request to the corresponding author. Sample queries should also be directed to the corresponding author.

**Supplement**

The Supplement related to this article is available online at: …

**Author contributions**

JEK designed the experiments, prepared the tables and figures and wrote the manuscript. JEK and LNS carried out the mechanical experiments and processed the data. JS prepared the samples and conducted physical measurements with AL, JEK and LNS. AFB, ODL, JEK and LNS processed the acoustic emission data. JEK, RC, TM and YL collected the samples. All authors contributed to the preparation of the manuscript.



**Competing interests**

The authors declare no competing interests.


**Acknowledgements**

The authors would like to thank Takeshi Matsushima and Hiroshi Shimizu for their support and guidance. JEK was funded by an Early Career Fellowship of the Leverhulme Trust (ECF-2016-325). LS and BK were supported by the Royal Society Te Apārangi Marsden project "Shaking magma to trigger volcanic eruptions" (15-UOC-049). YL, JS, AL, OL and RC

acknowledge funding from the European Research Council (ERC) Starting Grant on "Strain Localisation in Magma" (SLiM; no. 306488). Fieldwork was funded by the DAIWA Anglo-Japanese Foundation (grant number 11000/11740).

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
