# Peer review of "Physical and mechanical rock properties of a heterogeneous volcano; the case of Mount Unzen, Japan"

_Solid Earth, 2020_

## Referee Comment (RC1) · Michael Heap (Referee) · 27 Oct 2020

The submitted manuscript presents an impressive collection of petrophysical data for four blocks collected from Mt Unzen volcano in Japan. The manuscript is well organised, well written, and the vast (in terms of type and quantity) dataset presented will surely pique the interest of the volcano community. The theme of my main comments on the manuscript, see below, is that I think that the authors should add some caveats to some of their experimental data. I recommend publication after the following main comments and line-by-line comments (minor revisions) have been addressed to the satisfaction of the editor.

[Figure]

Mike Heap (University of Strasbourg)

Main comments

1. The volume of rock investigated by a hand-held permeameter depends strongly on the radius of the nozzle: the larger the radius of the nozzle, the larger the volume of rock investigated. Theory highlights that the zone of investigation is approximately four times the inner radius of the nozzle (see, for example, Goggin et al. (1986) "A Theoretical and Experimental Analysis of Minipermeameter Response Including Gas Slippage and High Velocity Flow Effects"). Given that the nozzle of the hand-held permeameter used in the submitted study has a radius of 4 mm, the minimum radius of a sample that can be measured is 16 mm. However, although some of the samples have a radius of 20 mm (the Brazil-disc samples), the majority of the prepared cores have a radius of 10 mm. Therefore, the 20 mm-diameter cores are too small to be measured by the hand-held permeameter. Further, if measurements were taken at multiple different positions on the flat end-face of these samples, the nozzle was likely positioned, at least for some of these measurements, very close to the edge of the sample. I'm not necessarily suggesting that the authors remove these data from the manuscript. However, if they are to remain in the manuscript, I strongly suggest that the authors clearly state that the hand-held permeameter measurements on the 20 mm-diameter samples do not conform with certain published theoretical guidelines and therefore very likely overestimate the permeability. Indeed, the cited Filomena et al. (2014) paper shows that measurements (nozzle diameter = 9 mm) made on 1-inch diameter cores are between 34 and 36% higher than those measured on the parent block. I therefore also strongly suggest that the authors use "permeability estimate" (this tool is designed for quick field assessments of permeability) rather than "permeability measurement" when referring to these data.

2. Related to the previous point, estimations of permeability using the hand-held permeameter on the laboratory samples will not be sensitive to permeability anisotropy. This explains why no permeability anisotropy was observed when using the handheld permeameter, but was observed when the permeability was measured in a pressure vessel using conventional techniques. If the authors choose to keep their handheld permeameter data, they should clearly state that assessments of permeability anisotropy on laboratory samples is not possible using the hand-held permeameter and it is for this reason why no permeability anisotropy is observed (Lines 477-482). Therefore, the discussion on Lines 732-736 and 747-763 should also be reduced.

3. Due to the high sample-to-sample variability of the blocks collected, robust assessments of water-weakening are not possible using only one wet uniaxial compressive strength or one wet indirect tensile strength measurement. The addition of wet UCS and tensile strength measurements is certainly a plus for the manuscript, but the authors should clearly inform the reader (in the results, discussion, and conclusions sections) that, due to sample-to-sample variability, robust assessments of water-weakening are not strictly possible with these data, but that pilot wet experiments are suggestive of water-weakening. Some of the discussion, which tries to explain why water-weakening is observed in compression but not in tension (starting on Line 790), should also probably be reduced. Unless the authors are willing/able to provide more wet experiments?

4. On Line 858 (and elsewhere) the authors state that "failure forecasting may be more effective in tensile regimes". Can the authors elaborate on what they mean here? Which parts of, or when is, a volcano typically in a tensile or a compressive regime? Is it possible to tell? Or, it is possible to infer the stress conditions from the recorded volcano seismicity?

Line-by-line comments

Line 83: Not only the porosity, but the connectivity of the porosity.

Line 136: I think this should be "...the strain of the material in response to loading..."

Line 136: I wouldn't say that Young's modulus correlates "poorly" with porosity. The

scatter is quite similar to that for UCS as a function of porosity ("Volcanic rock strength inversely correlates with porosity. . ."; Line 103).

Line 223: The authors previously state that there is "substantial hydrothermal alteration in localised areas of the dome" (Line 190). No hydrothermally altered rocks were collected during the field campaign? If hydrothermally altered rocks are not considered representative of the dome, perhaps the authors should provide some information as to the extent of the alteration? Is it restricted to only a small volume of the dome?

Line 331: How long did a typical experiment last?

Line 357: "The".

Line 522: "uniaxial".

Line 559: Is it useful, here or elsewhere, to compare this empirical relation with that presented in Heap et al. (2020; already cited) for a range of volcanic rocks?

Line 768: Missing space.

Line 796: For water-weakening in rocks, see Baud et al. (2000) "Failure mode and weakening effect of water on sandstone".

Line 838: Not only strength, Heap et al. (2020; already cited) show that Young's modulus is also considerably reduced when one takes large-scale fractures into account.

---

## Referee Comment (RC2) · Philip Benson (Referee) · 5 Nov 2020

This is very nice work, and a very interesting paper. The authors have collected a huge rock mechanics and rock physics dataset on volcanic rocks collected from Mt Unzen (Japan); the site of extensive activity in the early 1990's. It is always pleasant to see new datasets presented and published, as these data are hard to collect and will benefit the community for many years. I have only a few general comments, and some minor points for the authors to consider.

General comments:

[Figure]

1. How were the blocks selected in the field? Was this opportunistic, or were the sites selected via some form of criteria? This could, of course, be as simple as to cover a range of rock physical properties in 'accessible' locations, but it'd be nice to have that directly stated.

2. Somewhat covered by the earlier review: Permeability is easily one of the most tricky parameters to measure and discuss, particularly in the field, and in terms of spatial variation. Many years ago a NERC scheme (micro to macro) identified that permeability needed a measurement every few metres to see such variations, compared to 100's of metres and km to resolve parameters like elastic wave velocity and conductivity. That's just one example, but perhaps this type of 'challenge' is worth reinforcing when introducing and discussing the general nature of heterogeneity inherent in volcanic deposits of all kinds.

Minor queries:

3. Line 137: Typo, "ultransonic" should be 'ultrasonic'.

4. Line 141: For a recent report on damage and Vp changes in volcanic rocks see, for example: - Harnett, C.E., P.M. Benson, P. Rowley, and M. Fazio (2018), Fracture and damage localization in volcanic edifice rocks from El Hierro, Stromboli and Tenerife, Scientific Reports, 8, 1942, doi: 10.1038/s41598-018-20442-w.

5. Line 300: A pore pressure differential of 1.1 to 1.5 is actually fairly high considering the confining pressures of 5.5-13.5 MPa. Leading to what is, in effect, an 'effective pressure differential' across the length of the sample (rule of thumb being dP of around 10% of Pc, so 1.3MPa for the 13.5MPa experiment). Might the author comment or add a few words here? I suspect this protocol was adopted simply due to the low permeabilities of the rock types investigated, but it'd be good to have this confirmed by the authors.

6. Line 375: I wouldn't call this sub-section heading "Acoustic emissions - active":

surely you mean simply "elastic wave velocity"? Or perhaps "active surveys"? Rather a minor quibble, but I do think the use of AE is implied as passive only and this is well established in the literature.

7. Lines 695 (figure 8): What is the error on the velocity changes? Apologies if it is in the text, and I missed it when reading up to this point.

8. Lines 935-945, and a few other places: Do the authors note any differences in the character of the AE with regards to the dry and saturated experiments? This is a well known phenomenon in volcanic systems with the inherent fluid-rock coupling, for example: - Fazio, M., P.M. Benson and S.V. Vinciguerra (2017), On the generation mechanisms of fluid-driven seismic signals related to volcano-tectonics, Geophysical Research Letters, 44, 734-742, doi:10.1002/2016GL070919. - Fazio, M., Salvatore Alparone, Philip M. Benson, Andrea Cannata, Sergio Vinciguerra (2019), Genesis and mechanisms controlling Tornillo seismo-volcanic events in volcanic areas. Scientific Reports, 9, 7338, doi: 10.1038/s41598-019-43842-y I leave it to the authors as to whether they think it is worth including, or out of the scope of their study.

---

## Author Comment (AC1) · 17 Dec 2020

**Reviewer 1, Michael Heap**

The submitted manuscript presents an impressive collection of petrophysical data for four blocks collected from Mt Unzen volcano in Japan. The manuscript is well organised, well written, and the vast (in terms of type and quantity) dataset presented will surely pique the interest of the volcano community. The theme of my main comments on the manuscript, see below, is that I think that the authors should add some caveats to some of their experimental data. I recommend publication after the following main comments and line-by-line comments (minor revisions) have been addressed to the satisfaction of the editor.

We thank the reviewer, Dr. Heap, for their appraisal of the manuscript, and for raising some important comments which we have addressed in the revised version of the manuscript. Please see detailed answers below each of the reviewer comments.

Main comments

1. The volume of rock investigated by a hand-held permeameter depends strongly on the radius of the nozzle: the larger the radius of the nozzle, the larger the volume of rock investigated. Theory highlights that the zone of investigation is approximately four times the inner radius of the nozzle (see, for example, Goggin et al. (1986) "A Theoretical and Experimental Analysis of Minipermeameter Response Including Gas Slippage and High Velocity Flow Effects"). Given that the nozzle of the hand-held permeameter used in the submitted study has a radius of 4 mm, the minimum radius of a sample that can be measured is 16 mm. However, although some of the samples have a radius of 20 mm (the Brazil-disc samples), the majority of the prepared cores have a radius of 10 mm. Therefore, the 20 mm-diameter cores are too small to be measured by the hand-held permeameter. Further, if measurements were taken at multiple different positions on the flat end-face of these samples, the nozzle was likely positioned, at least for some of these measurements, very close to the edge of the sample. I'm not necessarily suggesting that the authors remove these data from the manuscript. However, if they are to remain in the manuscript, I strongly suggest that the authors clearly state that the hand-held permeameter measurements on the 20 mm-diameter samples do not conform with certain published theoretical guidelines and therefore very likely overestimate the permeability. Indeed, the cited Filomena et al. (2014) paper shows that measurements (nozzle diameter = 9 mm) made on 1-inch diameter cores are between 34 and 36% higher than those measured on the parent block. I therefore also strongly suggest that the authors use "permeability estimate" (this tool is designed for quick field assessments of permeability) rather than "permeability measurement" when referring to these data.

We thank the reviewer for this comment and indeed we recognise the limitations of the hand-held permeameter, especially in terms of its application to laboratory specimens of relatively small size. We're happy to change the terminology to "permeability estimate", and agree this is more accurate terminology, and have implemented this change in the revised manuscript (in some instances we still say "permeability" alone, where adding "estimate" would disrupt the flow of the sentence, and we refer to the physical act of |"making a measurement" where it has already been clearly established to which suite of measurements the text refers). We do feel that the measurements made using the handheld permeameter are an important part of the study, and as such we have opted to add reference in the method with regard to the limitations and theoretical guidelines that have been proposed for tinyperm measurements. In addition, realising that we have the dataset available to explore the potential impact of the sample size limitation further, we performed some further investigations. Recall, the core size was 20 mm diameter and 40 mm high, whilst disc size was 40 mm diameter and 20 mm high (with the discs being slightly over the minimum suggested radius for the handheld permeameter and the cores being below, according to Goggin et al., 1986), and for 2 blocks we also made measurements on planar block surfaces (of 80 x 180 mm for UNZ14 and 80 x 400 mm for UNZ9). We added reference to this further analysis in the method section 2.1.5.

Our further investigation first focuses on any differences betweeen measurements on cores and discs, which theoretically span the minimum size limit for minipermeameter measurements. In fig 4b of the manuscript we show each measurement made on each core and disc (overlain by the average for each sample chosen for the mechanical study). The data are also presented in Supplementary Table 1, and as such the reader (and reviewer) have access to all the data necessary to explore further. You can see in these data, and clearly in our plot below, that the permeability using the Tinyperm is very similar for both cores and discs (here each point represents the average from 5-10 measurements for a single disc or core, showing 59 cores and 55 discs), which plot largely indistinguishably in porosity-permeability space. If, as indicated by the recommended nozzle to sample radius of Goggin et al. (1996) the sampling volume were too small to be accurately measured in the cores, and yet acceptable in the discs, then one would expect the cores to plot at higher permeability (for example, Filomena et al. (2014) suggested an increase of 34-41% was seen when sample size was below the theoretical limit, and accordingly suggested a correction factor for Tinyperm measurements made on small specimens).

[Figure]

*N.b. fit lines are for visual guidance only*

This does not, however, rule out the possibility that both cores and discs (at ~12 and ~27 cm^3 respectively) are both below the size at which a representative permeability measurement can be made. So, to explore this aspect further we utilised data already collected, but collated it differently, we modified Table 2 in the main manuscript to address this comment. Here, we show a comparison of permeability measurements made on samples UNZ9 and UNZ14, on discs, cores and on planar block surfaces, showing the number of measurements, minimum, maximum, average, standard deviation, and coefficient of variation for the values at each different sample scale. We also added a plot below to summarise this data, which shows the minimum, maximum and average values recorded in each of the 3 geometry samples (blocks, cores and discs) for the two sample groups. What this data shows, is that although there are small differences in the minimum, maximum and average recorded in the samples of different size and geometry, that variation is not systematic with sample size. The averages in both samples (UNZ9 & 14) plot close together, whilst maximum and minimum values show greater disparity, and yet for sample UNZ14 the highest and lowest value is in the block sample and in UNZ9 the discs. Meanwhile, the coefficients of variation (which remove bias that might be seen in standard deviation due to the number of measurements made) show that both blocks have higher variability than either the cores or discs (see Table 2). These results indicate that there is no systematic difference in the measured permeability in the difference groups (and as such no underestimation of permeability) as a result of sample sample surface area (and ratio to nozzle size) or volume, across the range measured here, despite the fact that the cores fall below the theoretical limit of Goggin et al (1986), as mentioned by the reviewer.

[Figure]

Table 2 (from the manuscript):

| Sample | Geometry | Surface covered | Number of measurements | Permeability | | | | |
|---|---|---|---|---|---|---|---|---|
| | | | | Minimum | Maximum | Average | Standard deviation | Coefficient of variation |
| | | *cm²* | | *m²* | | | | *%* |
| UNZ9 | Block | 8 x 40 | 262 | 1.90E-15 | 1.51E-12 | 1.53E-13 | 2.19E-13 | 143.01 |
| | Cores | 2 (circular) | 210 | 3.73E-15 | 1.21E-12 | 2.20E-13 | 2.35E-13 | 107.15 |
| | Discs | 4 (circular) | 160 | 1.65E-15 | 2.18E-12 | 1.94E-13 | 2.71E-13 | 139.18 |
| UNZ14 | Block | 8 x 18 | 117 | 9.15E-15 | 2.58E-12 | 1.75E-13 | 2.85E-13 | 163.07 |
| | Cores | 2 (circular) | 100 | 3.72E-14 | 1.28E-12 | 2.64E-13 | 2.81E-13 | 106.14 |
| | Discs | 4 (circular) | 130 | 2.06E-14 | 5.06E-13 | 1.18E-13 | 9.75E-14 | 82.94 |

We posit a number of possible reasons why the sample size permeability discrepancy anticipated by former literature may not apply to our samples:

- It is possible that the relationship of Goggin et al. (1986) may not hold true for larger diameter nozzle contacts (note our nozzle size is 8 mm diameter, compared to their primary nozzle size of 0.625 mm diameter used in their study) though it is theoretically dimensionless.
- It is likely the case that absolute porosity (and permeability) also plays a role in the sensitivity of the handheld permeameter to sample size, though, we do not have the data here to explore such a relationship further as the connected porosity of the blocks we measured was very similar (13-16%; as they were chosen for their similar porosity but contrasting anisotropy). The Goggin et al. (1986) study focuses on lower porosity and permeability samples ($10^{12}$ to $10^{15}$ m²) than those studied here, and the Filomena et al (2014) study which reports a reduction in measured permeability moving from blocks to cores of 34-41 %, is based on limited measurements on samples of low (and relatively limited) porosity and permeability. A suggestion of the impact of porosity on the sample volume needed for an accurate measurement is in Fig 8 in Goggin et al (1986), where the offset between minipermeameter and hassler cell measurements is greater for the range 1-10 md (~$10^{14}$ to $10^{15}$ m²), which is approaching the lower end of capability for such devices anyway. We also wonder if Goggin et al.'s calibration measurements, which were made at > 2 MPa of effective pressure, contributed to the non-linear offset between minipermeameter and traditional hassler cell measurements as a function of porosity (Fig 8, Goggin et al., 1986), as it has been shown that effective pressure disproportionately reduces permeability of low porosity (already less permeable) samples. More work would be needed to conclude definitively if the need for a geometric correction is porosity-dependent.
- We do also believe the correction factor given by Filomena et al (2014) is too large. In the application of their own correction factor (defined by measurements on 4 blocks and 4 cores taken from the same blocks) to a broader dataset of 51 samples, Filomena et al. (2014) seem to demonstrate that their correction for the minipermeameter permeability is too significant: This is seen when comparing Tinyperm measurements to permeability measured using different approaches (in which the samples are jacketed) such as the Hassler cell (where the sample also has an acting effective pressure, which we know reduces permeability, e.g. fig 4c herein and many previous studies). In this demonstration (Table 1, Filomena et al., 2014) the permeability of 19 samples (from 51) is already lower with the TinyPerm than in the confined Hassler cell set-up (at 5MPa confining pressure – effective pressure not detailed), and after the correction factor (x 0.59-0.66) 36 samples' permeability measured by TinyPErm drop to lower values than those measured in the Hassler cell, which given the contrasting pressure conditions, is very unlikely to be accurate – the lack of confinement in the minipermeameter should consistently give higher permeabilities than the hassler cell for all samples. This demonstration that the correction to account for undersized samples appears to be too significant is further compounded in the offset between the different measurement types highlighted in Figs 5-6 of Filomena et al. (2014). This suggests that a correction that may be applicable to one, or limited, samples cannot be extrapolated to a broader variety of materials. We rather consider this as a demonstration that variability of tinyperm measurements may be relatively high - Lamur et al. (2019) attained between 0.2 (at low permeability) and 0.5-1 (at higher permeabilities) log units variability using repeat measurements in the same location on the same sample). Yet despite the variability also demonstrated herein, the values we obtain appear not to need correcting at the laboratory scale, or at least not at the scale and permeability range examined in this study, and rather such variability calls for an apporach which adopts many repeat measurements when using a minipermeameter, and if possible, a direct comparison to the same sample measured at larger scale (i.e. measuring the block prior to sample prep.) such as we used in this study to explore the need to correct data for scale. [Finally, we note that the 979 data points that comprise our further analysis is significantly higher than the number of measurements made in earlier studies which propose a correction may be necessary due to size limitations, and though we do not doubt that the effect was seen in those studies, in our study, we do not see an impact of the sample size.]

We thank the reviewer for drawing this additional use of our dataset to our attention, so in addition to giving the theoretical and previously observed limits in the method we now present the above in the results, and briefly discuss in section 4.1 (Relationships between physical and mechanical attributes) along with the modified text from comment 2 below. As mentioned we have modified Table 2 to include the new analysis of size/geometry impact, and we also added another table to the supplementary information containing each individual measurement on the block surfaces of UNZ9 and UNZ14 (as Table S2, and so we have shifted the numbering of other supplementary tables accordingly). Finally, in the previously submitted manuscript we compared the standard deviation and coefficient of variation measured on the block surfaces to those measured on the cores and discs, yet, this comparison was not strictly accurate as the core-disc values we previously showed were the averages from 5-10 measurements (see Table 1) – these values (of standard deviation and coefficient of variation) are valuable to look at variability from core to core (/ disc to disc), but should not be compared to the non-averaged standard deviation and coefficient of variation values from the block surfaces which consider each individual measurement. So, we now provide both the original values in Table 1 to look at repeatability from core to core (and disc to disc), and an additional measure of the mean, standard deviation, and coefficient of variation of all individual measurements made on the cores and discs, shown both in comparison to measurements on the blocks in modified Table 2 and with the complete permeability dataset presented in Table S1 – the corresponding text has been modified accordingly.

2. Related to the previous point, estimations of permeability using the hand-held permeameter on the laboratory samples will not be sensitive to permeability anisotropy. This explains why no permeability anisotropy was observed when using the hand-held permeameter, but was observed when the permeability was measured in a pres- sure vessel using conventional techniques. If the authors choose to keep their hand- held permeameter data, they should clearly state that assessments of permeability anisotropy on laboratory samples is not possible using the hand-held permeameter and it is for this reason why no permeability anisotropy is observed (Lines 477-482). Therefore, the discussion on Lines 732-736 and 747-763 should also be reduced.

We agree with the reviewer and admit that our observation of this fact (no anisotropy detected by the handheld permeameter) did not adequately express the current state of knowledge (that the measurement approach is not able to detect anisotropy) and nor did it fully capture our intended point which was that we expected higher variability of the values in the banded samples. We have thus, as suggested, shortened and better framed the discussion of the data in terms of the measurement approach (rather than sampling volume as it was primarily before).

3. Due to the high sample-to-sample variability of the blocks collected, robust assessments of water-weakening are not possible using only one wet uniaxial compressive strength or one wet indirect tensile strength measurement. The addition of wet UCS and tensile strength measurements is certainly a plus for the manuscript, but the authors should clearly inform the reader (in the results, discussion, and conclusions sections) that, due to sample-to-sample variability, robust assessments of water- weakening are not strictly possible with these data, but that pilot wet experiments are suggestive of water-weakening. Some of the discussion, which tries to explain why water-weakening is observed in compression but not in tension (starting on Line 790), should also probably be reduced. Unless the authors are willing/able to provide more wet experiments?

We agree with the reviewer that sample variability is relatively high in these samples, and as such the observed suggestion of water- weakening is not strictly statistically robust. That said, there are several aspects of the water saturated tests that are of particular importance to the study, hence our choice to include it: In particular the observation that water saturation can obscure the identification of changes in material damage by coda wave interferometry, and we thank the reviewer for noting these experiments are of benefit to the manuscript. We have been happy to add a caveat to the result and discussion sections with regard to interpreting differences on the basis of a single saturated test on each sample type. For UCS in particular other literature has reported the impact of water saturation on strength and Young's modulus. As such, here, we slightly reduced the discussion of water saturation on strength, proposing a slight reduction on compressive strength (as previous authors have noted similar observations we now try to draw in these observations and citations), and unsystematic impact on tensile strength. We do remain confident of the impact of water on reducing Young's modulus (which was the case for all the samples tested). In particular, we added the following text in the results:

"*We do however caution that relatively high variability is observed across the sample suite, and as such the saturated tests are only indicative of the impact water saturation may have on strength*."

"*The saturated UTS tests showed that 3 of the 5 sample groups had lower saturated tensile strength than the average of the dry tests, but 1 of the 5 was higher than any of the dry tests of their respective sample group (Fig. 5f-j, Table 4), indicating no systematic change in UTS under saturated conditions (Fig. 6b), though high sample variability may obscure the impact of water saturation on tensile strength*."

*"The Young's modulus was systematically reduced in all saturated compression tests (Fig. 6c, Table 4), though variability within sample groups was high at dry conditions."*

The changes in the discussion reflect this uncertainty of the impact of saturation and are more substantiative so we draw attention to the modified manuscript. In the conclusion the text has been modified to:

*"The impact of saturation on strength is inconclusive due to high sample variability, though it appears to decrease UCS and have an unsystematic impact on UTS."*

4. On Line 858 (and elsewhere) the authors state that "failure forecasting may be more effective in tensile regimes". Can the authors elaborate on what they mean here? Which parts of, or when is, a volcano typically in a tensile or a compressive regime? Is it possible to tell? Or, it is possible to infer the stress conditions from the recorded volcano seismicity?

For the time-being the authors remain tentative in the extrapolation of this observation to real-World deformation scenarios. For example, it should not be interpreted that this observation directly upscales into two different tectonic regimes, e.g. rift volcanism in tension versus continental collision in compression (though in some cases of volcanic instability it might translate fairly literally e.g. Acocella (2005)). Neither does it suggest volcanic systems in different regimes necessarily follow different patterns of seismic release (though that may well be true – e.g. it has been shown that seismic b-value is often elevated during volcanic eruptions compared to tectonic earthquakes, and so to imagine it varies systematically from one type of an eruption to another is not a big stretch!). Yet the result offers some hope as to the potential of real-time interpretation of changes in seismicity. e.g., if one is able to classify seismic events (into groups/families of events), one might be better equipped to examine the rate of acceleration of activity for each group, and if the focal mechanism is identified then perhaps a critical acceleration rate could be established for each regime – in other words, it might be possible to establish unique failure criteria informed by the real data, to interpret how activity might escalate or slow based on more tailored models for the specific regime. Such tailored approaches are likely to be possible with increasing computational efficiency. As a slight aside, the great majority of laboratory acoustic emission data has been gathered during compressive mechanical testing (confined or unconfined), whereas this study suggests that tensile regimes behave somewhat differently (so we should also consider them) - so the appearance of a critical rate of AE acceleration in one regime, may not be representative of imminent failure in another. We added a bit more detail to reflect these considerations to section 4.2, though we do not want to overstep the bounds of the study.

Line-by-line comments

Line 83: Not only the porosity, but the connectivity of the porosity.
Agreed and changed.

Line 136: I think this should be ". . .the strain of the material in response to loading. . ." Line 136: I wouldn't say that Young's modulus correlates "poorly" with porosity. The scatter is quite similar to that for UCS as a function of porosity ("Volcanic rock strength inversely correlates with porosity. . ."; Line 103).
This comment is in regard to the literature data from Heap et al. (2020). Agreed, we have rephrased to simply:
*"...Young's modulus indicates the stress-strain response to loading and correlates negatively with porosity (Heap et al., 2020 and references therein)."*

Line 223: The authors previously state that there is "substantial hydrothermal alteration in localised areas of the dome" (Line 190). No hydrothermally altered rocks were col- lected during the field campaign? If hydrothermally altered rocks are not considered representative of the dome, perhaps the authors should provide some information as to the extent of the alteration? Is it restricted to only a small volume of the dome?
We thank the reviewer for this comment. We mention on line 190 that "Ongoing fumarole activity and prolonged residence at elevated temperature has resulted in substantial hydrothermal alteration in localised areas of the dome (e.g. Almberg et al., 2008)". As stated, the alteration of the dome materials is a result of ongoing fumerole activity at the summit, as such the samples we collected in the block and ash flow deposits show no such alteration as they have not been subjected to alteration post-emplacement.

Although we do not test any altered samples in this study, in a previous study (Coats et al., 2018) we demonstrated that altered rocks collected from the lava dome were mechanically largely indistinguishable from the pristine rocks collected from block and ash flow deposits. This was already mentioned elsewhere in the manuscript, but not directly addressed in terms of our sample selection. Here, rather than tackling the impact of alteration we instead focus on primary textural distinctions between samples, since porosity has a dominant control on strength, and anisotropy in

sheared samples is important in the Unzen lava dome (which consists of 13 distinct lobes and is highly sheared in places e.g. Wallace et al., 2019). We note that alteration would likely impact each of these materials differently due to the accessibility of the void space, and we agree this would make an interesting future study, though sampling of the lava dome is strictly limited in order to preserve it. In the current manuscript, we now added some text to clarify the targets of our sample choice in the method (as a result of this comment, and a comment from reviewer 2 on sample selection) and present the observation that alteration minimally impacts strength from Coats et al (2018) in the introduction.

Line 331: How long did a typical experiment last?

Brazil tests took between 205-430 seconds, so conform to ASTM standard in terms of time to failure (1-10 min), though they are slower than ISRM (15-30 s) – as the recommendations do not match, and as one of our goals was to look at temporal evolution of the materials during stressing it suited us to opt for the longer timeframe afforded by ASTM guidelines. We added a sentence to this effect in the method section 2.2.2.

Line 357: "The".
removed capitalisation

Line 522: "uniaxial".
removed capitalisation

Line 559: Is it useful, here or elsewhere, to compare this empirical relation with that presented in Heap et al. (2020; already cited) for a range of volcanic rocks?

We thank the reviewer for this comment, though we find a discussion of the empirical model of Heap et al., 2020, fits better in the interpretation section 4.1 (Relationships between physical and mechanical attributes). We added a brief discussion of our relationship given in Eq. 12 in comparison to the power law and exponential empirical relationships of Heap et al. (2020):

*"Young's modulus of the materials tested mimicked patterns observed for strength, showing a non-linear negative correlation with porosity which is described by Eq. (12). This relationship shows a slightly lower dependence of Young's Modulus on porosity than previously described exponential and power law relationships developed for a range of volcanic materials (Heap et al., 2020), which we attribute to: 1) the lower span of porosity used to define our relationship (9-38 % compared to 3 – 50 %), in particular the lack of low porosity samples, for which Young's Modulus rises increasingly non-linearly; 2) the relatively high prevalence of fractures in our samples (as compared to more equant vesicles in some volcanic rocks); and 3) the inclusion of anisotropic cataclastic samples with porous and dense bands"*

The plot below shows a comparison of our fit in comparison to those from Heap et al. (2020):

[Figure]

Line 768: Missing space.
Added space after "(21.38 MPa)"

Line 796: For water-weakening in rocks, see Baud et al. (2000) "Failure mode and weakening effect of water on sandstone".

We have edited this part of the discussion due to an earlier comment, so now cite a number of additional literature examples of water weakening in different lithologies, including the suggested paper by Baud et al. 2000.

Line 838: Not only strength, Heap et al. (2020; already cited) show that Young's modulus is also considerably reduced when one takes large-scale fractures into account.

Thanks for pointing this out, we added reference to this relevant finding: *"...upscaling of mechanical properties may see intact rock strength values further weakened by as much as 80-97 % (Thomas et al., 2004; Walter et al., 2019) and Young's Modulus reduced by up to a factor of 4 depending on the Geological Strength Index of the rock mass (see Heap et al., 2020 and references therein)."*

---

## Author Comment (AC2) · 17 Dec 2020

**Reviewer 2, Phil Benson**

This is very nice work, and a very interesting paper. The authors have collected a huge rock mechanics and rock physics dataset on volcanic rocks collected from Mt Unzen (Japan); the site of extensive activity in the early 1990's. It is always pleasant to see new datasets presented and published, as these data are hard to collect and will benefit the community for many years. I have only a few general comments, and some minor points for the authors to consider.
We thank the reviewer, Dr. Benson, for his synopsis and comments, which are addressed individually below.

General comments:
1. How were the blocks selected in the field? Was this opportunistic, or were the sites selected via some form of criteria? This could, of course, be as simple as to cover a range of rock physical properties in 'accessible' locations, but it'd be nice to have that directly stated.
We thank the reviewer for the comment. The answer is rather complicated, but in short, yes, we aimed to select a range of porosity blocks which were both isotropic and anisotropic. We actually selected around 14 blocks to bring back to the UK for a range of distinct studies. In the field these were selected from block and ash flow deposits due to accessibility, initially for favourable size and shape (to ensure we could get enough samples for repeats but also that we could carry and ship them) as well as a couple of criteria, for example we wanted them to appear uniform across the whole block (no large porosity contrast, etc) and not have any block-scale damage like through-going fractures that would impact ability to prepare samples. Once we had selected a number of samples, we examined them to compare to what we had seen during our time in the field (prior to sampling we had been working on the lava dome), and we also made some basic density estimates and measured some samples using tinyperm (though this was not very consistent as weather was poor and measurements do not work on wet surfaces). We then tried to fill any gaps in our sampling, knowing we ought to represent approx. 1.5-2.4 g/cm$^3$ density (see Kueppers et al., 2005- https://doi.org/10.1016/j.jvolgeores.2004.09.005). For example it was difficult to locate mid-porosity samples (20-25 %) which did not bear a fabric, though this might represent a slight bimodal distribution which has been previously observed at Unzen (Kueppers et al., 2005). We also knew that much of the dome contains sheared magmas (e.g. Wallace et al., 2019), so selected the block with visible shear textures on the surface. When we were able to bring the samples into the lab we cut each to see the internal texture, prepared a slice for thin section and a core to measure porosity. After that we selected which samples could be used for each study (e.g. this study, and Coats et al., 2018, etc.) and compared to our previously selected samples (e.g. Hornby et al., 2015). Because the purpose of the present study was to observe the impact of porosity across the spectrum of materials, we selected samples that spanned the density range measured at Unzen previously (Kueppers et al., 2005, compared to our Supplementary Table 1), and used the sample with cataclastic banding collected specifically for the study to compare the role of anisotropy. Our existing description in section 2.1.1. sample collection, has been modified to describe our sampling rationale:

*"During a field campaign in 2015 a suite of blocks, each > 15 kg were collected from block-and-ash flow deposits on the eastern and north-eastern flanks (Fig. 1b). The samples were assessed in the field to ensure representative texture and estimated densities that matched the known range of physical attributes of Unzen lavas (cf. Kueppers et al. 2005). The target was to select 4 blocks for this study which spanned low (UNZ14), medium (UNZ1) and high (UNZ13) porosity, plus an additional block that displayed an anisotropic cataclastic fabric (UNZ9) as the summit lava dome is pierced by shear zones (see, e.g. Wallace et al., 2019). Blocks UNZ1 and UNZ13 were also used for the study by Coats et al. (2018) which examined the role of temperature, alteration and strain rate on the rheological response to deformation at high temperature and defined a failure criterion for porous dome rocks and lavas."*

2. Somewhat covered by the earlier review: Permeability is easily one of the most tricky parameters to measure and discuss, particularly in the field, and in terms of spatial variation. Many years ago a NERC scheme (micro to macro) identified that permeability needed a measurement every few metres to see such variations, compared to 100's of metres and km to resolve parameters like elastic wave velocity and conductivity. That's just one example, but perhaps this type of 'challenge' is worth reinforcing when introducing and discussing the general nature of heterogeneity inherent in volcanic deposits of all kinds.
We thank the reviewer for this suggestion, we tried not to venture too far beyond the bounds of our dataset, since we do not tackle upscaling directly, yet we wanted to stress the importance for the volcanological community to 1) use real values and data collected and 2) develop approaches that can actually upscale these values in a meaningful way. We modified some of the discussions around permeability due the comments of reviewer 1 also, and we made sure to mention the difficulties of upscaling permeability, which is a vital parameter in interpreting volcanic unrest. We really need, as a community, to tackle upscaling of permeability in a meaningful way.

Minor queries:
3. Line 137: Typo, "ultransonic" should be 'ultrasonic'.

Thanks, changed.

4. Line 14: For a recent report on damage and Vp changes in volcanic rocks see, for example: - Harnett, C.E., P.M. Benson, P. Rowley, and M. Fazio (2018), Fracture and damage localization in volcanic edifice rocks from El Hierro, Stromboli and Tenerife, Scientific Reports, 8, 1942, doi: 10.1038/s41598-018-20442-w.
We thank the reviewer for this suggestion, which we have now cited this relevant manuscript in a number of places, in both introduction and discussion.

5. Line 300: A pore pressure differential of 1.1 to 1.5 is actually fairly high considering the confining pressures of 5.5-13.5 MPa. Leading to what is, in effect, an 'effective pressure differential' across the length of the sample (rule of thumb being dP of around 10% of Pc, so 1.3MPa for the 13.5MPa experiment). Might the author comment or add a few words here? I suspect this protocol was adopted simply due to the low permeabilities of the rock types investigated, but it'd be good to have this confirmed by the authors.
We apologise that this was actually imprecision in our description of the methodology. We actually varied the flow rate until the inlet and outlet pressure stabilised, we targeted an outlet pressure between 1.1-1.5 MPa (which we chose as a target – this was the reference to the setpoint in the previous version of the method), at which point we locked the pressure of both inlet and outlet (i.e. set the pressure differential) and let the flow rate equilibrate again before taking the permeability measurement. This is a slightly unusual approach, chosen as the samples span quite a range of permeabilities. The average differential pressure of our measurements was actually 0.28 MPa (not 1.1 to 1.5 MPa as was wrongly insinuated, which was the outlet pressure), all the pressure and flow rate data for each test are shown in Supplementary Table S3 (previously S2). We thank the reviewer for highlighting the confusing description of the methodology, which we have now rectified in section 2.1.6 (Confined water permeability).

6. Line 375: I wouldn't call this sub-section heading "Acoustic emissions - active": surely you mean simply "elastic wave velocity"? Or perhaps "active surveys"? Rather a minor quibble, but I do think the use of AE is implied as passive only and this is well established in the literature.
We agree that "active surveys" is a good title, our original terminology was due to wanting to clarify the distinction between section 2.2.4 and 2.2.5 but is not needed, so we have changed it.

7. Lines 695 (figure 8): What is the error on the velocity changes? Apologies if it is in the text, and I missed it when reading up to this point.
This is a good question, and not one we can fully quantify, unfortunately. The acoustic emissions we recorded were at KHz, so it might be reasonable to say that there was an error of at least 0.001%. However, we expect it could be higher due to potential misalignment of the pulses during the stacking stage. Even misalignment of 1 sample would introduce errors, and because tests are noisy this is possible. So, we anticipate maybe the error would be on the order of 0.01% velocity change, i.e. approx. 1 % of the measurement. As this is speculative, we added a comment as to the accuracy of the values in the discussion (as opposed to the method which states the sampling rate).

8. Lines 935-945, and a few other places: Do the authors note any differences in the character of the AE with regards to the dry and saturated experiments? This is a well known phenomenon in volcanic systems with the inherent fluid-rock coupling, for example: - Fazio, M., P.M. Benson and S.V. Vinciguerra (2017), On the generation mechanisms of fluid-driven seismic signals related to volcano-tectonics, Geophysical Research Letters, 44, 734-742, doi:10.1002/2016GL070919. - Fazio, M., Salvatore Alparone, Philip M. Benson, Andrea Cannata, Sergio Vinciguerra (2019), Genesis and mechanisms controlling Tornillo seismo-volcanic events in volcanic areas. Scientific Reports, 9, 7338, doi: 10.1038/s41598-019-43842-y I leave it to the authors as to whether they think it is worth including, or out of the scope of their study.
We thank the reviewer for the comment and we also would very much like to explore this further. However, we have not been able to make this distinction, unfortunately - due to the active surveys we had a lot of noise in our wet samples, and we were not able to reliably exclude the survey pulses from the passive AE recording, hence our omission of passive AE in the saturated tests in the manuscript (note we were able to extract enough pules with confidence to track the velocity, but not all of them so as to reliably trust the passively recorded data).